Subject Areas:
behaviour/ecology

Keywords:
elasmobranchs, bioacoustics, behaviour, southern stingray, threshold, frequency

Author for correspondence:
Megan F. Mickle
e-mail: micklem@uwindsor.ca

# Field assessment of behavioural responses of southern stingrays (*Hypanus americanus*) to acoustic stimuli

Megan F. Mickle, Rachel H. Pieniazek and Dennis M. Higgs

Department of Biological Sciences, University of Windsor, 401 Sunset Avenue, Windsor, Ontario Canada, N9B 3P4

MFM, 0000-0002-9872-1986; DMH, 0000-0002-0771-4642

The ability of elasmobranchs to detect and use sound cues has been heavily debated in previous research and has only recently received revived attention. To properly understand the importance of sound to elasmobranchs, assessing their responses to acoustic stimuli in a field setting is vital. Here, we establish a behavioural audiogram of free-swimming male and female southern stingrays (*Hypanus americanus*) exposed to low-frequency tones. We demonstrate that female stingrays exposed to tones (50–500 Hz) exhibit significant changes in swimming behaviours (increased time spent swimming, decreased rest time, increased surface breaches and increased side swimming with pectoral flapping) at 140 dB re 1 µPa ($-2.08$ to $-2.40$ dB re 1 m s$^{-2}$) while males exposed to the same tones did not exhibit a change in these behaviours until 160 dB re 1 µPa ($-1.13$ to $-1.21$ dB re 1 m s$^{-2}$). Our results are the first demonstration of field responses to sound in the Batoidea and show a distinct sensitivity to low-frequency acoustic inputs.

## 1. Introduction

Elasmobranchs possess multiple sensory and behavioural adaptations used for communication and migration, however, less is known about their hearing, mechanosensory systems and functional use of the auditory system compared to teleosts [1]. Evidence suggests that elasmobranchs, e.g. lemon sharks (*Negaprion brevirostris*) and Atlantic stingrays (*Hypanus sabinus*), use their inner ears and lateral line to orient themselves to biotic sounds, such as prey items [2–8], suggesting that sound detection may be important to the overall

fitness of the animal. The hearing range of elasmobranchs studied to date falls within the range of 20–1000 Hz with greatest sensitivities at lower frequencies [3,9–12]. Elasmobranch and teleost ears are both comprised of inner ear labyrinths containing a saccule, lagena, utricle, otoliths and three semicircular canals; however, the elasmobranch ear is unique in that it also contains the macula neglecta, which, combined with the sacculus, is thought to be used for hearing [13–15]. Elasmobranchs detect sound through particle motion (not pressure) because they lack a swim bladder and specialized hearing structures [4,9,16–19] which typically act as pressure-to-displacement transducer organs in teleosts.

There is some evidence that elasmobranchs show an attraction response to low-frequency pulsed sounds [6,20–22], as they are thought to mimic stimuli produced by struggling prey [7]. Playback experiments using pulsed sounds have been found to attract over 20 species of sharks, some from up to 100 m from the initial sound source [14], although there has been criticism of this earlier work because sharks should only detect particle motion, which would not transmit far from the source [12,19,23]. While there are several attraction experiments performed in elasmobranchs there is limited data available regarding their potential adverse responses to sound. Klimley & Myrberg [24] performed one of the few experiments establishing an avoidance response of sharks to sound in an aquarium setting, demonstrating that lemon sharks (*Negaprion brevirostris*) withdraw when presented with a broad band sound with a sudden onset of high intensity sound (20 dB above the ambient level). Recently, Chapius *et al.* [25] showed that killer whale (*Orcinus orca*) calls and artificial sounds composed of mixed tones at frequencies from 20 Hz to 10 kHz caused a decrease in approaches of eight species of reef and coastal sharks to a baited camera set-up. However, Ryan *et al.* [26] showed no effect of sound alone as a deterrent to feeding in Port Jackson (*Heterodontus portusjacksoni*) and epaulette (*Hemiscyllium ocellatum*) sharks in a laboratory setting. Therefore, the mixed results of sound as a deterrent further show the need for enhanced field tests of behavioural responses of elasmobranchs to acoustic stimuli.

Elasmobranchs face several anthropogenic stressors, such as habitat degradation [27], overfishing/ bycatch [28–30] and provisioning [31,32], and are threatened in all the world's oceans [33,34]. Rising levels of anthropogenic sounds are also increasingly recognized as a global threat to fishes [35,36], yet there are few studies conducted to date focusing on potential anthropogenic sound threats on elasmobranchs, and of the few studies, most focus on sharks with batoids poorly represented [37]. Here, we exposed male and female southern stingrays (*Hypanus americanus*) to low-frequency tones (50–1000 Hz) at differing sound levels to create a behavioural audiogram of these animals. We show significant differences in male and female responses to sound, with females responding to lower levels compared to males. This data can be further used to make hypotheses on potential noise impacts on stingrays.

# 2. Methods

## 2.1. Capture and transportation

Experiments were conducted following Canadian Council for Animal Care (CCAC) protocols (University of Windsor AUPP 17–11). Experiments took place at the Bimini Biological Field Station (BBFS) on the small island of South Bimini, Bahamas, which has minimum commercial boating activity and some recreational power boats. Our study species was the southern stingray which is found in western Atlantic coastal waters, is a common benthic mesopredator, has a diet composed mostly of crustaceans and teleosts, and exhibits sexual dimorphism where females on average are larger than males [38–41]. To catch stingrays, two 16-foot Sundance skiffs (Mercury Sea Pro engine), equipped with two rubber dip nets (40″ × 40″), four spoons (devices made of two PVCs and plastic netting) and a plastic holding pool (approx. 4 × 4 × 2 ft), were driven to the mangroves around the South Island of Bimini, during February–March 2019. Stingrays were often caught during mid-tide as this was the most efficient time to catch the animals. Once an animal was spotted, individuals (approx. 6 in total) got into the water to surround and capture the animal and transfer it to the holding tank on the skiff with the dip net. Once captured, animals were scanned using a PITtag reader (GPR Plus Reader, Biomark); if the stingray was previously experimented on, it was released. If the animal did not have a PITtag number, they would be tagged following experimentation to avoid further stress to the animal prior to the study. Stingrays were then transported to a holding net pen (15 × 10 m) kept in the ocean at BBFS, with total travel times between 5 and 18 min from capture. Animals acclimated in the holding pen for approximately 24–40 h prior to experimentation. Stingrays were then identified again using the PITtag reader and transported from the holding pen to the experiment pen. Male stingrays caught ranged in total length from 44.6 to 115.2 cm while females ranged from 107 to 140 cm; there was no exclusion of stingrays caught based on their size.

**Table 1.** Sound level measured as acceleration units (dB re 1 m s$^{-2}$). Acceleration was estimated by the pressure gradient between hydrophone readings taken exactly 1 m apart using the Euler equation [46].

| frequency (Hz) | decibel level (dB re 1 µPa) | | | | |
|---|---|---|---|---|---|
| | 130 | 135 | 140 | 150 | 160 |
| 50 | −2.93292 | −2.47048083 | −2.40206 | −1.61007 | −1.20703 |
| 90 | −2.75886 | −2.44377776 | −2.11007 | −1.46007 | −1.21509 |
| 200 | −2.85886 | −2.37562758 | −2.12494 | −1.59378 | −1.07494 |
| 500 | −2.70703 | −2.31006671 | −2.08504 | −1.35169 | −1.1283 |
| 1000 | −2.75703 | −2.41508854 | −2.07563 | −1.85703 | −1.12563 |

## 2.2. Experiment

A circular experimental net pen (5 × 5 m), composed of metal rebar, plastic mesh netting extending underneath the sand, and two GoPro cameras (Hero 7) mounted to the rebar, was created in the ocean to establish a behavioural audiogram of stingrays in both a control and noisy setting. During experimentation, outside temperatures ranged from 17 to 30°C, water temperature ranged from 22 to 23°C, water depth ranged from 35 to 75 cm and wave conditions ranged from 0 to 2 on the Beaufort scale. Quantified behaviours included: time spent swimming, resting, side swimming (time spent swimming vertically along the perimeter of the pen flapping pectoral fins) and surface breaches (head out of the water). As stingrays are generally sedentary animals [40,42] and settled to the bottom after acclimation in our experiments, an increase in swimming behaviour along the bottom of the pen was used as the prime metric for a threshold response to sound and increases in side swimming and breaching behaviours were used as metrics of an agitated response to the sound stimuli.

To perform sound experiments, two low-frequency underwater speakers (Clark Synthesis Diluvio AQ339; Lubell Labs) were placed adjacently along the perimeter of the pen, connected to an amplifier (Scosche SA300), a 12 volt PBS car battery, and an mp3 player (Sony Walkman NWZ-E464). Twenty stingrays (nine males and 11 females) were exposed to five low-frequency tones: 50, 90, 200, 500 and 1000 Hz, for these tones are hypothesized to overlap with their hearing range [12]. The sequence of each frequency was randomized using a random number generator application to avoid pseudoreplication. Testing involved a 2–3 h acclimation period in the experimental net pen followed by sounds played in a stepwise pattern with a 1 min sound period followed by 5 min of silence until all sounds were presented [43–45]. Stingrays were tested individually to avoid any follower bias. Behaviours during the 1 min sound treatment were directly compared with the 1 min of silence prior to sound, creating a difference metric. Sound level was measured in pressure units (dB re 1 µPa) using a hydrophone (Inter Ocean system inc. – Acoustic Calibration and System Model 902), as well as acceleration units (dB re 1 m s$^{-2}$). Acceleration was estimated by the pressure gradient between hydrophone readings taken exactly 1 m apart using the Euler equation ([46], table 1). While we recognize that acceleration units are the most relevant for detection at the level of the ray ear [9,17–19], we also provide pressure units to make it easier for other investigators measuring sound in the ocean, as underwater accelerometers remain difficult to obtain for open-field studies [47,48]. A sound map was created to measure background sound levels along 27 locations of the net-pen using a hydrophone (Inter Ocean system inc. – Acoustic Calibration and System Model 902). The range of decibel levels for each frequency were also measured along 27 locations of the pen to establish a range of sound intensity the animals were exposed to during experimentation (figure 1 and table 2). Background sound levels were also measured in the middle of the pen prior to each experiment and ranged from 113 to 124 dB re 1 µPa. Each tone (50–1000 Hz) was also recorded underwater using a hydrophone (Loggerhead Instruments, Model # HTI-96-Min/3 V/Exp/LED) to display a visual representation of each frequency (figure 2). To establish the auditory threshold of stingrays, both males and females were first exposed to each frequency at 140 dB re 1 µPa (−2 dB re 1 m s$^{-2}$, table 2). If the animal exhibited a change in swimming level, then decibel levels were decreased (by 5 dB) until the animal stopped exhibiting a response to the sound. If the animal did not alter their movement at 140 dB re 1 µPa, decibel levels were increased (by 10 dB) until a change in behaviour was noted. Decibel levels were increased by 10 instead of 5 dB as it was found that increases by increments of 5 dB re 1 µPa showed no change in behaviour. However, it is noteworthy to mention that there were three male stingrays exposed to 150 dB initially, due to time constraints based on decreasing tide. To help ensure animals were not responding to the speakers' baseline electrical output, we also played

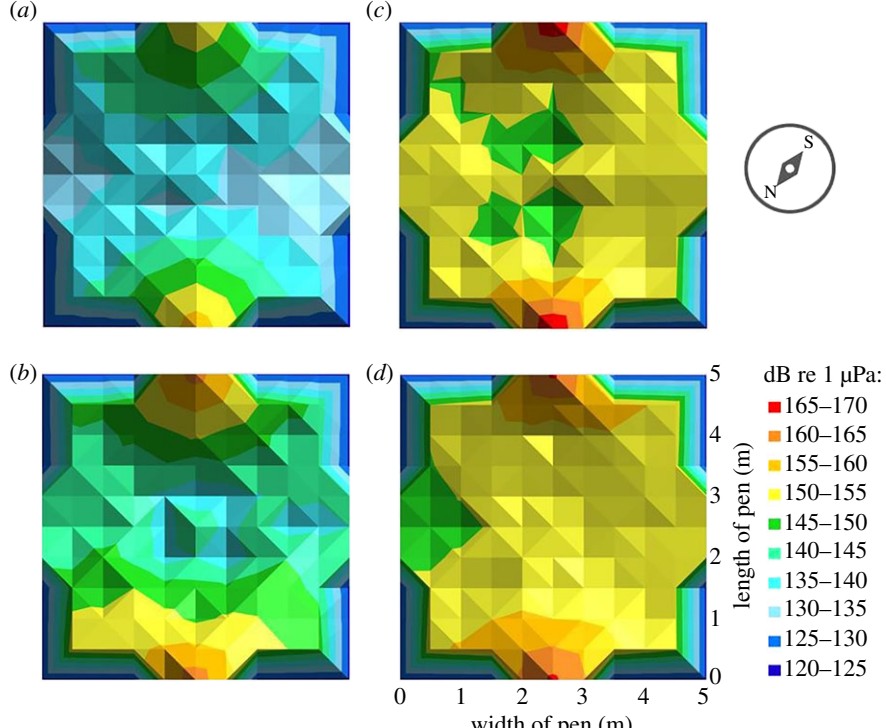

**Figure 1.** A sound map showing sound levels (dB 1 μPa) across the experimental pen for sound presentations of (*a*) 50 Hz at 140 dB re 1 μPa, (*b*) 500 Hz at 140 dB re 1 μPa, (*c*) 50 Hz at 160 dB re 1 μPa, (*d,b*) 500 Hz at 160 dB re 1 μPa. Sound level was measured at 27 locations in the pen, with intermediate levels interpolated between recording locations to represent sound level as a heat map across the entire experimental arena.

**Table 2.** Average decibel levels of each frequency along 27 locations (figure 1) of the net-pen to establish a range of sound intensity the animals were exposed to during experimentation.

| decibel levels (dB re 1 μPa) | 50 Hz | 90 Hz | 200 Hz | 500 Hz | 1000 Hz |
|---|---|---|---|---|---|
| 130 | 131.2 | 130.2 | 130.7 | 129.8 | 129.5 |
| 135 | 134.6 | 136.7 | 135.4 | 134.2 | 136.1 |
| 140 | 137.3 | 141.5 | 143.1 | 143.3 | 140.2 |
| 150 | 147.5 | 145.1 | 146.3 | 149.3 | 148.3 |
| 160 | 156.1 | 157.8 | 155.5 | 156.3 | 153.4 |

10 000 Hz at 150 dB re 1 μPa (−1.6903 dB re 1 m s$^{-2}$) to each animal, although we do recognize they may have also responded to the speaker electrical output driven by the lower frequency sound output (i.e. a 100 Hz sound stimulus may also output a 100 Hz electrical output in addition to the speaker background output). After each experiment, stingrays were sexed and measurements of disc length, disc width, spiracle width and total length were taken (table 3). Fieldwork presents difficulties in terms of weather conditions, so to best control for this, experimental trials were performed on days with similar weather, temperature conditions and tide range. Behaviour videos were analysed using the software program 'Soloman Coder' (Version: beta 19.08.02).

## 2.3. Statistical analysis

Prior to the field season a power analysis was conducted to determine an appropriate sample size (eight stingrays of each sex) and to ensure the assumptions of normality would not be violated. Differences in stingray behaviour between the control and treatment periods were assessed using repeated measures ANOVA for each metric: time spent swimming or resting, side swimming and surface breaches. To account for habituation or potential differences due to time spent in pen, behaviours were analysed as a difference metric between the final minute of the 5-min control and the 1-min sound presentation. To create the difference metric, simple contrasts relative to 0 in the repeated measures design were

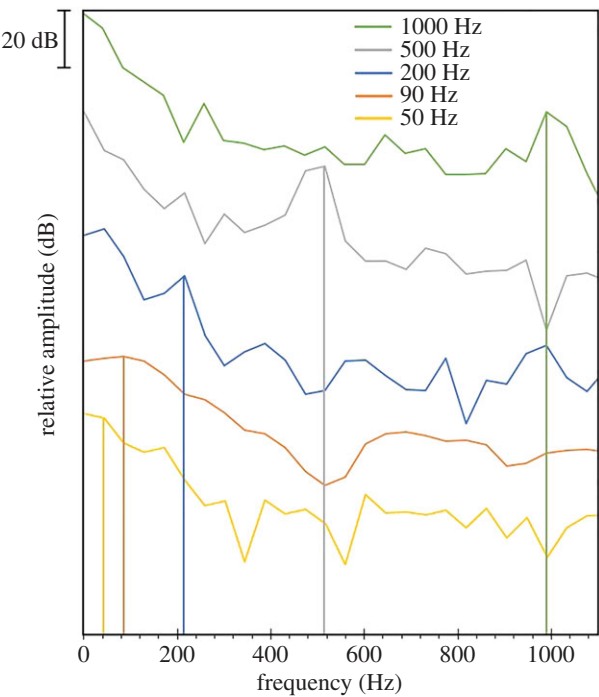

**Figure 2.** A visual representation of the frequency range (50, 90, 200, 500 and 1000 Hz) and relative amplitude (dB) of each tone played to the stingrays during experimentation. All traces are at the same amplitude scale (see 20 dB scale bar) but are separated for clarity. Recordings were taken with a hydrophone (loggerhead Instruments, Model # HTI-96-Min/3 V/Exp/LED).

**Table 3.** The range of measurements of both male and female stingrays used in experiments.

| measurements (cm) | female | male |
|---|---|---|
| total length | 107–140 | 44.6–115.2 |
| disc width | 59.2–91 | 42.9–55.2 |
| disc length | 48–74 | 35.7–43.4 |
| spiracle width | 9–15.2 | 6.9–9.9 |
| barb length | 9.8–10.8 | 5.5–9.9 |

implemented. A linear regression was performed to determine if the size of the stingray and water depth influenced swimming activity of the fish. The regression was performed at the determined average threshold of the stingrays, 140 dB re 1 µPa for females and 160 dB re 1 µPa for males, at 50 Hz, as any potential influence of water depth would be greatest at the lowest frequency presented. Normality was tested using the descriptive statistics function in SPSS (IBM SPSS Statistics).

# 3. Results

Both male and female southern stingrays exhibited a change in behaviour when exposed to low-frequency tones. Females responded to tones at an average RMS of 140 dB re 1 µPa, while males responded to an average RMS of 160 dB re 1 µPa (see table 1 for respective acceleration units throughout results).

## 3.1. Threshold differences

We used swimming activity along the bottom of the pen as a marker of hearing to determine threshold. For females, there was an overall significant effect of sound level on swimming activity at 50, 90, 200 and 1000 Hz (table 4 and figure 3*a*). In subsequent *post hoc* analysis at these frequencies, there was no difference in swimming activity of female stingrays when comparing 130 and 135 dB re 1 µPa ($p > 0.999$ for all frequencies, figure 3*a*) but there was an increase in swimming activity at 140 dB re 1 µPa relative to the lower sound levels across all frequencies ($p < 0.004$, figure 3*a*).

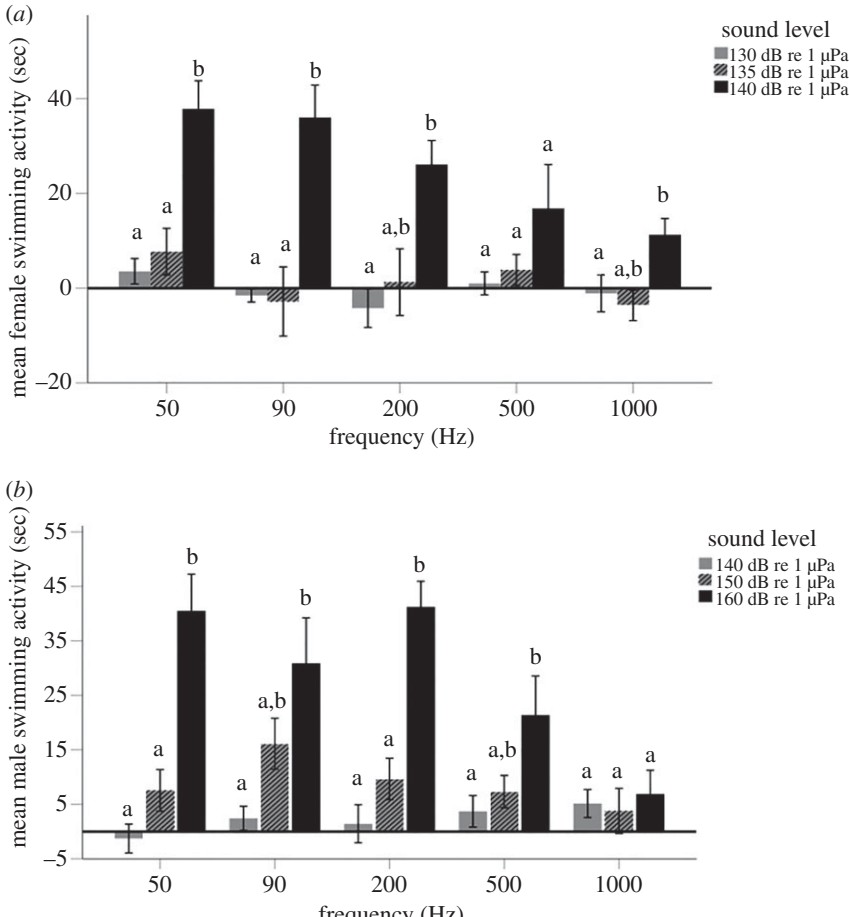

**Figure 3.** The hearing threshold of (*a*) female and (*b*) male southern stingrays using swimming activity as a marker of hearing. Significant differences between the sound levels are indicated by different letters. Error bars are representative of mean (+/− s.e.).

**Table 4.** Statistical representation of main effects of frequencies (50–1000 Hz) at 140 dB re 1 µPa for females and 160 dB re 1 µPa for males, on swimming activity levels.

| frequency (Hz) | female | | male | |
| --- | --- | --- | --- | --- |
| | *F* statistic | *p*-value | *F* statistic | *p*-value |
| 50 | $F_{3,29} = 10.803$ | $p < 0.001$ | $F_{2,21} = 20.331$ | $p < 0.001$ |
| 90 | $F_{3,29} = 9.218$ | $p < 0.001$ | $F_{2,21} = 5.604$ | $p = 0.011$ |
| 200 | $F_{3,29} = 5.684$ | $p = 0.003$ | $F_{2,21} = 26.052$ | $p < 0.001$ |
| 500 | $F_{3,29} = 3.826$ | $p = 0.0120$ | $F_{2,21} = 3.673$ | $p = 0.043$ |
| 1000 | $F_{3,29} = 1.392$ | $p = 0.265$ | $F_{2,21} = 0.165$ | $p = 0.849$ |

There was an overall significant effect of sound level on swimming activity in males at 50, 90, 200 Hz (table 4 and figure 3*b*). *Post hoc* analyses indicate no difference in swimming activity of male stingrays between 140 and 150 dB re 1 µPa ($p > 0.968$ for all frequencies, figure 3*b*) but there was a significant difference between 140 and 160 dB re 1 µPa, as males exhibited an increase in swimming activity at 50, 90, 200 Hz at 160 dB re 1 µPa ($p > 0.008$, figure 3*b*). Males also displayed an increase in swimming activity at 50 and 200 Hz at 160 dB re 1 µPa when compared with 150 dB re 1 µPa ($p > 0.001$, figure 3*b*).

## 3.2. Frequency differences

To examine a behavioural response to frequency, activity, resting and side swimming as well as number of surface breach events were measured at 140 dB re 1 uPa for females and 160 dB re 1 uPa for males, as

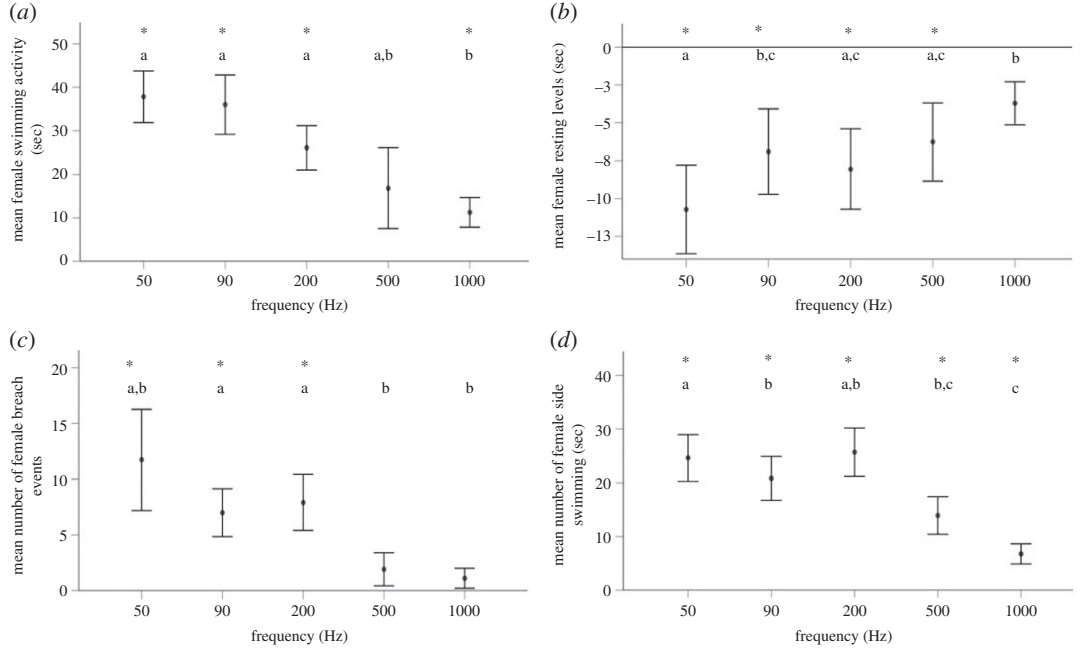

**Figure 4.** Female stingray behaviour in response to low-frequency tones at 140 dB re 1 µPa, relative to controls (silent period between sound) (*a*) mean swimming activity, (*b*) mean resting activity, (*c*) mean surface breach events and (*d*) mean time spent side swimming. Error bars are representative of mean (+/− s.e). Significant differences relative to control are indicated by an * while differences compared to other frequencies are indicated by different letters.

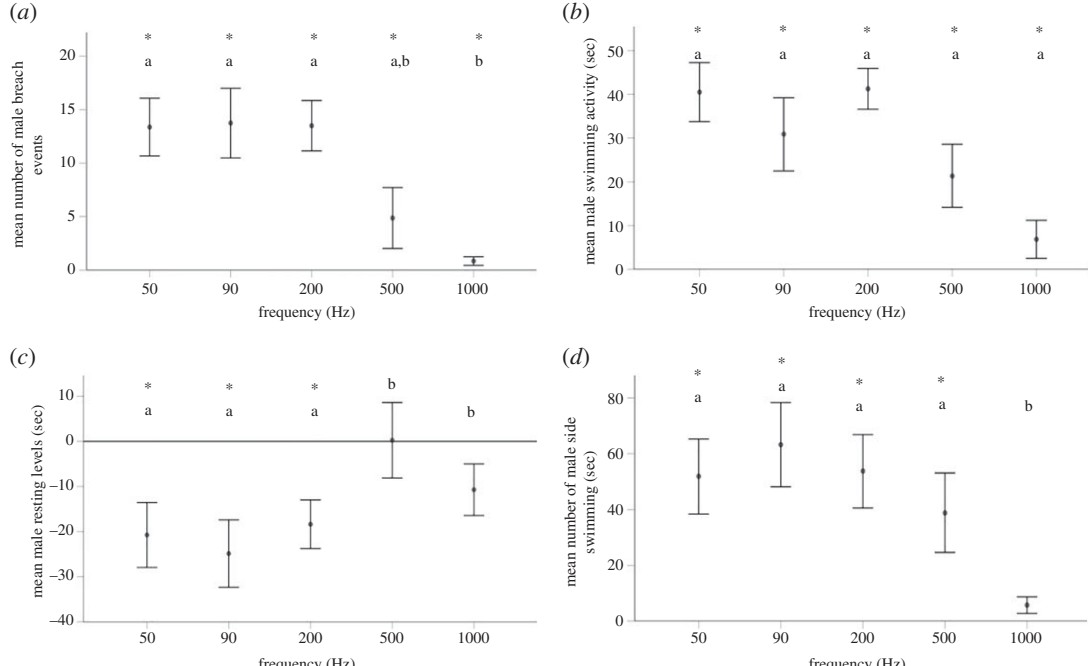

**Figure 5.** Male stingray behaviour in response to low-frequency tones at 160 dB re 1 µPa, relative to controls (silent period between sound). (*a*) Mean swimming activity, (*b*) mean resting activity, (*c*) mean surface breach events and (*d*) mean time side swimming. Error bars are representative of mean (+/− s.e.). Significant differences relative to control are indicated by an * while differences compared to other frequencies are indicated by different letters.

these were both identified as threshold intensities for each sex. There was a significant effect of frequency on the swimming activity of both female ($F_{5,6}$ = 10.935, $p$ = 0.006, figure 4*a*) and male stingrays ($F_{5,3}$ = 16.470, $p$ = 0.022, figure 5*a*). *Post hoc* analyses demonstrated significant increases in swimming activity of females relative to the control period at 50, 90, 200 and 1000 Hz while males were affected by 50, 90, 200 and 500 Hz. The number of surface breach events was also affected by sound exposure for

both females ($F_{5,5} = 1.932$, $p \leq 0.001$, figure 5$c$) and males ($F_{5,2} = 16.578$, $p \leq 0.012$, figure 5$c$) with 50, 90 and 200 Hz tones showing a significant increase in breaching events compared to controls. Time spent resting in females ($F_{5,5} = 3.291$, $p \leq 0.019$, figure 4$b$) and males ($F_{5,2} = 2.226$, $p = 0.050$, figure 5$b$) was also affected by frequency, causing a decrease in female resting rates when exposed to 50, 90, 200 and 500 Hz and male resting rates at 50, 90 and 200 Hz. Side swimming increased during sound exposure for both females ($F_{5,6} = 7.237$, $p \leq 0.030$, figure 4$d$) and males ($F_{5,3} = 3.418$, $p \leq 0.029$, figure 5$d$) at 50, 90, 200 and 500 Hz. Neither male (rest: $F_{1,8} =$, $p = 0.347$; breach: $F_{1,8} = 1.00$, $p = 0.347$; swimming: $F_{1,6} = 2.141$, $p = 0.194$; side swimming: $F_{1,8} = 0.885$, $p = 0.374$) nor female (rest: $F_{1,10} = 0.820$, $p = 0.170$; breach: $F_{1,10} = 1.00$, $p = 0.341$; swimming: $F_{1,10} = 0.188$, $p = 0.674$; side swimming: $F_{1,10} = 1.815$, $p = 0.208$) stingrays responded to 10 000 Hz indicating that behavioural changes were true acoustic responses and not responses to speaker electrical output. The variations of water depth and total fish length did not have a significant effect on the activity levels of male (adjusted $R^2 = -0.138$; $F_{1,7} = 0.151$, $p = 0.711$; adjusted $R^2 = 0.192$; $F_{1,7} = 2.428$, $p = 0.180$, respectively) and female (adjusted $R^2 = -0.029$; $F_{1,10} = 0.721$, $p = 0.418$; adjusted $R^2 = -0.045$, $F_{1,10} = 0.566$, $p = 0.471$, respectively) stingrays when exposed to 50 Hz at their average thresholds. Behaviour data were normally distributed.

# 4. Discussion

For the first time, we quantify the behavioural thresholds of southern stingrays to a sound source and demonstrate that females respond to lower decibel levels (140 dB re 1 µPa; $-2$ dB re 1 m s$^{-2}$) than males (160 dB re 1 µPa; $-1$ to $-1.2$ dB re 1 m s$^{-2}$).

## 4.1. Threshold

Southern stingrays are generally sedentary bottom-dwelling animals [40,42], therefore, we quantified resting behaviour as residing at the bottom of the pen without movement, while an increase in swimming along the bottom of the pen indicated a response to sound. Stingrays were haphazardly distributed along the southeast quadrant of the net pen facing the open ocean (figure 2) prior to experimentation, indicating that stingrays were exposed to similar sound levels at the start of each experiment. As stingrays increased time spent swimming and decreased resting time during treatments ranging from 50 to 500 Hz, we conclude that sound elicited a change in normal stingray behaviour. While we did note that females still exhibited a response at 1000 Hz, which was not detected in males, the swimming activity that was recorded was less than in the 50, 90, 200 and 500 Hz.

Barber *et al*. [49] discovered significant sex differences in the macula neglecta and ramus neglectus of the thornback ray (*Raja clavata*), with hair cell and axon number increasing with the size of the animal, and females have larger hair cell counts compared to similarly sized males. As previously mentioned, the macula neglecta and sacculus are used primarily for hearing, therefore gender differences may be involved in the location of prey, mate detection, or other reproductive processes [50]. Corwin *et al*. [51] discovered that the number of hair cells in *R. clavata* increases from 500 at birth to approximately 6000 in 7-year-old rays, further hypothesizing that the increase in hair cell counts is related to an increase in hearing sensitivity. For the current study, female stingrays had an average total length of 141.5 cm while the males were on average 88.9 cm (table 3). As hair cell numbers in the macula neglecta increase with the size of the animal, the differences in hearing threshold observed in the current study may simply be explained by the size of macula neglecta and number of hair cells, however, we did not see a significant effect of the size of the stingray on swimming activity levels of both male and females. Additionally, the size of the male and female rays were similar to those in the study by Barber *et al*. [50] and there was still a significant difference in hair cell and axon number in the macula neglecta. Therefore, more research is needed to determine if differences are based on age, size, or sex of the animals. Future studies should focus on similarly sized male and female stingrays to determine if sex differences in hearing threshold are still present.

## 4.2. Frequency

Our findings on frequency detection are consistent with previous research showing that elasmobranchs can hear from 20 to 1000 Hz with greatest sensitivities at lower frequencies [3,9–12]. Our behavioural evidence suggests that southern stingrays are most sensitive to frequencies of 50 up to 500 Hz, which is consistent with the results obtained from Corwin, demonstrating that the best hearing sensitives of the thornback ray (*Raja clavata*) are between 40 and 200 Hz.

When exposed to lower frequencies, stingrays exhibited an increase in surface breach behaviour and while there is little data to explain this behaviour in stingrays, some animals jump or escape the water as a means of avoiding stressful conditions or to escape predators [45,52,53]. Therefore, we also hypothesize that stingrays breach the water to avoid sounds.

## 4.3. Future considerations and conclusions

For the first time, we show behavioural evidence of hearing in a stingray species, confirming that rays hear within the range of 50–1000 Hz, with greatest sensitives at lower frequencies and that females respond to lower sound levels than males. Recently, laboratory-based hearing studies have come under increasing scrutiny due to problematic acoustics in tank environments [48,54,55], suggesting an increased need for studies in a more natural setting. However, conducting acoustic studies in a field setting presents some challenges as field conditions cannot be as readily controlled as in a laboratory environment. For example, there was a variation in depth (35 to 75 cm), weather conditions (cloud cover) and wave action (0–2 on a beaufort scale) during experimentation. As wild stingrays were caught for experimentation, they exhibited a variation in size, and there is no way of knowing if they were exposed to anthropogenic sound prior to capture. Furthermore, the stingrays may have detected the frequencies at lower intensities of sound and did not respond, but without using an auditory evoked potential (AEP) technique, we cannot successfully examine this phenomenon. While underwater acoustics are never perfect, and we still see a use for laboratory-based experiments, our approach offers a promising avenue for continued investigation of auditory responses in elasmobranchs.

The behavioural responses that were observed to low-frequency sounds, especially an increase in surface breach behaviour, raises concerns regarding the exposure of elasmobranchs to anthropogenic sounds. The increasing anthropogenic cacophony of the underwater environment is of significant concern worldwide and has already been shown to have important fitness consequences for teleosts [35,56–58]. Despite the concerns for anthropogenic sound as an ecological stressor, we have no knowledge of these possible effects on elasmobranchs, one of the most imperiled groups of fish worldwide [27,33]. Next steps should include quantifying the effects of sound as a potential stressor on southern stingrays and expanding the current approach to other elasmobranchs.

Ethics. All work regarding the use of live animals has been approved by the Canadian Council on Animal Care (CCAC) under the AUPP 17–11.
Data accessibility. Raw data are presented in an electronic supplementary material excel file submitted to Royal Society Open Science.
Authors' contributions. M.M. project design and development, data collection and analysis, manuscript writing and editing. R.P. data collection, field assistant, manuscript editing and D.H. project design and development, manuscript editing, funding support.
Competing interests. We declare we have no competing interests.
Funding. This research was funded by the Natural Sciences and Engineering Research Council (NSERC).
Acknowledgements. We thank Kirsten Poling, Christina Semeniuk and Nigel Hussey for assistance with designing this project. We also thank Bimini Biological Field Station for assistance with stingray capturing and data collection.

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
