## [Reviewer comments · Royal Society Open Science]

Review History

RSOS-191544.R0 (Original submission)

Review form: Reviewer 1

Is the manuscript scientifically sound in its present form?

No

Are the interpretations and conclusions justified by the results?

No

Is the language acceptable?

Yes

Do you have any ethical concerns with this paper?

No

Have you any concerns about statistical analyses in this paper?

No

Recommendation?

Major revision is needed (please make suggestions in comments)

Comments to the Author(s)

I was very interested to read your manuscript about the auditory sensitivity of southern stingrays and was especially pleased to see immediately in your abstract the links of particle motion as well as sound pressure, which is often missing in the field.

However, after reading the manuscript, I am very concerned about the title of this paper.

Behavioural audiograms are a controversial field in underwater acoustics as the “true” hearing ability of the individuals being tested may be obscured by a habituation of an individual to the sound exposure.

Furthermore, I am not completely convinced that your differences are purely related to sex, they may be linked to age, prior exposure etc.

Specific comments

Abstract

Within the abstract you note “clear changes in swimming behaviours”, it would be nice if this early on in the manuscript you are able to explicitly state whether the behaviours were “significant”.

You also subtly use hyperbole in your abstract, suggesting that 140dB is a low sound level. This is a matter of to whom is listening, as from an acoustics point of view 140dB is pretty loud and represents anthropogenic activity in the field.

I am concerned with the ending of the abstract, and that you are suggesting that the swimming behaviour response is directly linked to hearing ability. Even at this early stage as a reader I think that there could be differences in age of the individuals, in sensitivity to sound from prior experience. So it is very difficult to make the assumption that there is a difference in males versus females.

Introduction

Opening paragraph

In the opening paragraph of the introduction, you use the terminology ears. I think you should be using inner ear.

Can you include which groups, species of elasmobranchs have been studied by the many references you cite, instead of just listing many.

Second paragraph

When you state “to which stimuli” you should include, particle motion or sound pressure I would refrain from using the word “noise” in the manuscript as this is very subjective.

Were the studies looking at the effect of anthropogenic sound done in lab tanks or the field? This may have had a large influence owing to the low frequency component being tested.

Rather than “and colleagues” use “et al.”

Do not use common names such as “orca”, include genus species and the common name used globally killer whale. Furthermore, what were the artificial sounds used in this study (pure tones, broadband, sound level, distance away). Try not to be vague in such statements.

Third paragraph

Again, you are vague by using “clear difference”, making me question whether it is actually significant to be in a scientific paper?

The closing sentence of the introduction is very confusing. Are you going to be looking at juveniles, adults, did you raise stingrays in tanks to insure no prior exposure. It also seems to be very out of place, as up until now you have not mentioned the effect of age on hearing ability.

Methods

Capture and transportation

Were the individuals caught of similar size (males on size category) (females one size category)?

Yes, you did not experiment on an animal twice. However, I am concerned that there is no way of knowing if the individual being tested was subject to high intensity anthropogenic sound prior to

being caught and hence may have auditory damage. Which undeniably is a very unrepresented area of bioacoustics.

Experiment

You state the measurements of the net pen, but do not specify the depth of the water column where the net pen was placed. Did the nets reach the ocean floor?

Acceleration was measured using the hydrophone. How did you ensure neutral buoyancy as well as factor in wave motion/ current motion associated with open water conditions?

I found it interesting that you started off the behavioural experiments with the highest sound level (140 – 150 dB). I think this is a flaw in the methodology as it could be suggested that you have “informed” the animal that an experiment is happening which could either dampen or accelerate further responses.

What were the range of temperature conditions as you did not include this information like waves and depth variation?

Also were you using the Beaufort scale to qualitatively justify low wave conditions?

Results

The opening statement of the results is that females responded at 140 dB and males 160dB. Is there average? RMS level SPL? Is this the average/ max/ min across the population used?

Why was swimming activity activity used as a marker for hearing and not any of the other metrics?

The results section is quite repetitive. I wonder if the statistical results would be better given in a table.

I think individual measurements need to be highlight for those performed in the lowest tidal range and highest tidal range (35 – 75cm), as well as highlighting the largest and smallest individuals to show that this wasn't a factor in differences recorded.

Discussion

Threshold

In the second paragraph you state that “there is little evidence” but then go on to give several examples of evidence. Therefore, I would suggest omitting this statement.

You begin to elude that the difference seen in your results may be as a result of size of individuals.

I think it might not just be size but also age of individuals. Is there or was there any way to age the individuals used in this study?

Frequency

You mention the advantages of using your “net pen” compared to tank environment studies.

However, I think you should also include the disadvantages of your study. Variable depth being a massive factor and concern as a reader.

Future considerations

Does the habitat of the southern stingray directly overlap with commercial/ recreational activity. If so, how “loud” and frequent is it in the area.

Tables

Table 1: Why a larger disparity in the size of females. What time of year is the reproductive season? Does this coincide with your collection time?

Figures

Figure 5: I do not understand this figure at all. Which being from an underwater acoustics background is surprising to me. What am I looking at on the y axis? Why are the figures not in order.

Figure 1: You need to plot a soundmap on top of your image of the locations to show where there were acoustic anomalies or hotspots within the tank. Throughout the paper you also have no mention of the distribution of the stingrays in the tank at the offset of sound exposure. You have video footage that you used for analysis so I think plotting start and stop times for different individuals as well as patterns of movement would not be a stretch.

Figures 2 – 4 need ticks on the x and y axis.

Review form: Reviewer 2

Is the manuscript scientifically sound in its present form?

Yes

Are the interpretations and conclusions justified by the results?

Yes

Is the language acceptable?

Yes

Do you have any ethical concerns with this paper?

No

Have you any concerns about statistical analyses in this paper?

Yes

Recommendation?

Major revision is needed (please make suggestions in comments)

Comments to the Author(s)

See attached (Appendix A).

Review form: Reviewer 3

Is the manuscript scientifically sound in its present form?

Yes

Are the interpretations and conclusions justified by the results?

No

Is the language acceptable?

Yes

Do you have any ethical concerns with this paper?

No

Have you any concerns about statistical analyses in this paper?

No

Recommendation?

Major revision is needed (please make suggestions in comments)

Comments to the Author(s)

I applaud the authors for undertaking a very challenging experiment. Very little is known about elasmobranch hearing and very few acoustical aquatic studies are done using behavior in the field. These are difficult experiments and really needed to understand hearing under quasi natural conditions. The reporting of both sound pressure and particle acceleration significantly strengthens the paper. The change in swimming behavior due to sound in rays is novel and would be of interest to the bioacoustics community.

I have several issues that may result from the very short descriptions provided in the manuscript. Additional details may alleviate some of my concerns. Also why is this manuscript single spaced with new line numbers every page? This is a strange format.

Major issues

I am confused about the apparent lack of controls in this study. What was the result when a stingray was captured, put in behavioral pen without sound exposure. Perhaps some of the behavior that the authors noted was an artifact of recent capture and handling. Having controls that were not exposed to sound would seem to be a necessary requirement for this study. The author need to be exceedingly careful what they conclude. It is a very good study but some of the conclusions are not supported. What the authors found where that larger female sting rays behaved differently in response to sound than smaller male stingrays. This does not mean that the sexes have different hearing sensitivities, it only means that they have different behavior responses to sound and it may be due solely to the size differences as the authors discuss. The authors need to be careful with other conclusions. For example, the following sentence "We uncovered a differential response in hearing sensitivity of male and females, discovering that females are more sensitive to sound level" .

Overstates the data.

This is again seen

"Now that we have demonstrated the baseline hearing ability of the southern stingray "

The authors have demonstrated what is the minimum sound to cause extra swimming in the ray. While I realize they are trying to argue that behavior experiments show greater sensitivity than AEPs, they cannot determine if the ray detected lower intensity sounds and did not respond. It is similar to teleost predator prey studies. While you can determine the distance that fish reacted to the prey, you cannot be certain, it did not detect the prey at greater range but only reacted at shorter range to increase capture success. Considering, the authors indicate that the sound is "bothering" the rays, then one can imagine that "sub bothering" thresholds could result in detectiob but not in motor activity. I would suggest substituting a more scientific description than "bother"

"We also present clear evidence that the rays were bothered by increasing sound levels. An escape response was quantified when the stingray swam along the perimeter of the pen while flapping their pectoral fins; as this is not a common behaviour exhibited by wild stingrays, we hypothesize that the animals are trying to avoid or escape the sound. When exposed to lower frequencies, stingrays exhibited an increase in surface breach behaviour. While there is little data to explain this behaviour in stingrays, some animals jump or escape the water as a means of avoiding stressful conditions or to escape predators (Baylis 1982; Gibb et al. 2011; Mickle et al., 2018). Therefore, we also hypothesize that stingrays breach the water to avoid sounds"

The above paragraph is purely speculative and is not supported by data. Exactly what does "bothering" mean. While the authors did a credible job indicated that the stingrays increase their swimming during sound presentation, the other information in this paragraph is not supported. The rays are in an confined settings. Interactions with the perimeter or perhaps electric currents due to the presence of the rebar in the marine environment were not controlled for and cannot be discounted. True escape behaviors localize and move away from the stimulus and the experiment is not set up to test this. I routinely see pectoral fin flapping on the surface during sustained, straight line swimming events in Florida with rays when there is no evidence of a predator.

In the methods: the behavior was compared to one minute of “silence” prior to the sound turned on. What happened after the sound was turned off? Was the behavior sustained? Was five minutes a sufficient interval for the ray to regain normal behavior? Did they habituate to the sound? Perhaps they were irritated throughout the study resulting in greater movement. It is critical that the authors demonstrate that after the sound was stopped that the rays showed control behavior.

Minor

Opening sentence of abstract is strange. Most aquatic animals are hypothesized to detect sound over great distances because sound travels great distances. However, much of the shark work is focused on long range olfactory detection. I would suggest a more focused opening statement that does not detract from your study

Was there not a recent Australian study that concluded sound was not effective as a deterrent in sharks, this should be discussed

Perhaps this species has different behavior, but the coastal rays I have observed in Florida have extensive swimming bouts including pectoral fins out of the water. There was no predator in the water. Also, escape response usually mean M-cell mediated escape response in teleosts. Need to make this clear.

Are the tables formatted correctly. Seems an excessive number of vertical and horizontal lines
Figure 1: A diagram depicting a sound map of the experimental setup. Decibel levels were measured every meter along 4 transecting lines (including edges).

There must be something wrong with my figure as it not a sound map. It is a plain ellipse and the distances do not appear to be to scale. For example, the horizontal midline is 3.5 m on the left and ~2.5m on the right.

I strongly suggest that the authors show a representative sound map for the study

Figure 2: The hearing sensitivity of a) female and b) male southern stingrays using swimming activity as a marker of hearing. Significant differences between the sound levels are indicated by different letters. Error bars are representative of mean (+/- S.E)

Again, although the authors define hearing, it is not true hearing sensitivity. Rather it is the threshold that induces additional swimming. While I understand that the authors did not report true controls, they should at least compare this values to the 1 min control they did use

Fig 3 and 4

I would like to see the control values plotted. Also are not swimming and resting reciprocal? Therefore, resting may not need to be reported

(silent period between noise). I thought that only 1 min was being compared in this period

Perhaps this journal has really strange formatting rules, but Figure 3 and 4 legends look like multiple choice exam questions rather than professional figure legends

Decision letter (RSOS-191544.R0)

04-Nov-2019

Dear Ms Mickle,

The editors assigned to your paper ("Sex differences in hearing sensitivities of southern stingrays (*Hypanus americanus*)") have now received comments from reviewers. We would like you to revise your paper in accordance with the referee and Associate Editor suggestions which can be found below (not including confidential reports to the Editor). Please note this decision does not guarantee eventual acceptance.

Please submit a copy of your revised paper before 27-Nov-2019. Please note that the revision deadline will expire at 00.00am on this date. If we do not hear from you within this time then it will be assumed that the paper has been withdrawn. In exceptional circumstances, extensions may be possible if agreed with the Editorial Office in advance. We do not allow multiple rounds of revision so we urge you to make every effort to fully address all of the comments at this stage. If deemed necessary by the Editors, your manuscript will be sent back to one or more of the original reviewers for assessment. If the original reviewers are not available, we may invite new reviewers.

- Data accessibility

If you wish to submit your supporting data or code to Dryad (<http://datadryad.org/>), or modify your current submission to dryad, please use the following link:
<http://datadryad.org/submit?journalID=RSOS&manu=RSOS-191544>

- **Competing interests**

- **Authors' contributions**

- **Acknowledgements**

- **Funding statement**

Kind regards,
Andrew Dunn
Senior Publishing Editor
Royal Society Open Science
openscience@royalsociety.org

on behalf of Dr Ryan Y Wong (Associate Editor) and Kevin Padian (Subject Editor)
openscience@royalsociety.org

Associate Editor's comments (Dr Ryan Y Wong):

Associate Editor: 1

Comments to the Author:

Dear Megan Mickle,

Your manuscript has now been seen by 3 reviewers. Their reviews are included. While all reviewers find the study interesting and a useful contribution, a number of concerns were raised.

Of particular note are the concerns regarding appropriateness of terminologies used (e.g. "behavioral audiogram"), insufficient methodological details, and the need for more cautious interpretations of the data. I recommend a major revision before considering the manuscript further.

Comments to Author:

Reviewers' Comments to Author:

Reviewer: 1

Comments to the Author(s)

I was very interested to read your manuscript about the auditory sensitivity of southern stingrays and was especially pleased to see immediately in your abstract the links of particle motion as well as sound pressure, which is often missing in the field.

However, after reading the manuscript, I am very concerned about the title of this paper.

Behavioural audiograms are a controversial field in underwater acoustics as the "true" hearing ability of the individuals being tested may be obscured by a habituation of an individual to the sound exposure.

Furthermore, I am not completely convinced that your differences are purely related to sex, they may be linked to age, prior exposure etc.

Specific comments

Abstract

Within the abstract you note "clear changes in swimming behaviours", it would be nice if this early on in the manuscript you are able to explicitly state whether the behaviours were "significant".

You also subtly use hyperbole in your abstract, suggesting that 140dB is a low sound level. This is a matter of to whom is listening, as from an acoustics point of view 140dB is pretty loud and represents anthropogenic activity in the field.

I am concerned with the ending of the abstract, and that you are suggesting that the swimming behaviour response is directly linked to hearing ability. Even at this early stage as a reader I think that there could be differences in age of the individuals, in sensitivity to sound from prior experience. So it is very difficult to make the assumption that there is a difference in males versus females.

Introduction

Opening paragraph

In the opening paragraph of the introduction, you use the terminology ears. I think you should be using inner ear.

Can you include which groups, species of elasmobranchs have been studied by the many references you cite, instead of just listing many.

Second paragraph

When you state "to which stimuli" you should include, particle motion or sound pressure

I would refrain from using the word "noise" in the manuscript as this is very subjective.

Were the studies looking at the effect of anthropogenic sound done in lab tanks or the field? This may have had a large influence owing to the low frequency component being tested.

Rather than "and colleagues" use "et al."

Do not use common names such as "orca", include genus species and the common name used globally killer whale. Furthermore, what were the artificial sounds used in this study (pure tones, broadband, sound level, distance away). Try not to be vague in such statements.

Third paragraph

Again, you are vague by using "clear difference", making me question whether it is actually significant to be in a scientific paper?

The closing sentence of the introduction is very confusing. Are you going to be looking at juveniles, adults, did you raise stingrays in tanks to insure no prior exposure. It also seems to be very out of place, as up until now you have not mentioned the effect of age on hearing ability.

Methods

Capture and transportation

Were the individuals caught of similar size (males on size category) (females one size category)?

Yes, you did not experiment on an animal twice. However, I am concerned that there is no way of knowing if the individual being tested was subject to high intensity anthropogenic sound prior to being caught and hence may have auditory damage. Which undeniably is a very unrepresented area of bioacoustics.

Experiment

You state the measurements of the net pen, but do not specify the depth of the water column where the net pen was placed. Did the nets reach the ocean floor?

Acceleration was measured using the hydrophone. How did you ensure neutral buoyancy as well as factor in wave motion/ current motion associated with open water conditions?

I found it interesting that you started off the behavioural experiments with the highest sound level (140 – 150 dB). I think this is a flaw in the methodology as it could be suggested that you have “informed” the animal that an experiment is happening which could either dampen or accelerate further responses.

What were the range of temperature conditions as you did not include this information like waves and depth variation?

Also were you using the Beaufort scale to qualitatively justify low wave conditions?

Results

The opening statement of the results is that females responded at 140 dB and males 160dB. Is there average? RMS level SPL? Is this the average/ max/ min across the population used?

Why was swimming activity activity used as a marker for hearing and not any of the other metrics?

The results section is quite repetitive. I wonder if the statistical results would be better given in a table.

I think individual measurements need to be highlight for those performed in the lowest tidal range and highest tidal range (35 – 75cm), as well as highlighting the largest and smallest individuals to show that this wasn't a factor in differences recorded.

Discussion

Threshold

In the second paragraph you state that “there is little evidence” but then go on to give several examples of evidence. Therefore, I would suggest omitting this statement.

You begin to elude that the difference seen in your results may be as a result of size of individuals.

I think it might not just be size but also age of individuals. Is there or was there any way to age the individuals used in this study?

Frequency

You mention the advantages of using your “net pen” compared to tank environment studies.

However, I think you should also include the disadvantages of your study. Variable depth being a massive factor and concern as a reader.

Future considerations

Does the habitat of the southern stingray directly overlap with commercial/ recreational activity. If so, how “loud” and frequent is it in the area.

Tables

Table 1: Why a larger disparity in the size of females. What time of year is the reproductive season? Does this coincide with your collection time?

Figures

Figure 5: I do not understand this figure at all. Which being from an underwater acoustics background is surprising to me. What am I looking at on the y axis? Why are the figures not in order.

Figure 1: You need to plot a soundmap on top of your image of the locations to show where there were acoustic anomalies or hotspots within the tank. Throughout the paper you also have no mention of the distribution of the stingrays in the tank at the offset of sound exposure. You have video footage that you used for analysis so I think plotting start and stop times for different individuals as well as patterns of movement would not be a stretch.

Figures 2 - 4 need ticks on the x and y axis.

Reviewer: 2

Comments to the Author(s)

See attached

Reviewer: 3

Comments to the Author(s)

I applaud the authors for undertaking a very challenging experiment. Very little is known about elasmobranch hearing and very few acoustical aquatic studies are done using behavior in the field. These are difficult experiments and really needed to understand hearing under quasi natural conditions. The reporting of both sound pressure and particle acceleration significantly strengthens the paper. The change in swimming behavior due to sound in rays is novel and would be of interest to the bioacoustics community.

I have several issues that may result from the very short descriptions provided in the manuscript. Additional details may alleviate some of my concerns. Also why is this manuscript single spaced with new line numbers every page? This is a strange format.

Major issues

I am confused about the apparent lack of controls in this study. What was the result when a stingray was captured, put in behavioral pen without sound exposure. Perhaps some of the behavior that the authors noted was an artifact of recent capture and handling. Having controls that were not exposed to sound would seem to be a necessary requirement for this study. The author need to be exceedingly careful what they conclude. It is a very good study but some of the conclusions are not supported. What the authors found where that larger female sting rays behaved differently in response to sound than smaller male stingrays. This does not mean that the sexes have different hearing sensitivities, it only means that they have different behavior responses to sound and it may be due solely to the size differences as the authors discuss. The authors need to be careful with other conclusions. For example, the following sentence "We uncovered a differential response in hearing sensitivity of male and females, discovering that females are more sensitive to sound level" .

Overstates the data.

This is again seen

"Now that we have demonstrated the baseline hearing ability of the southern stingray "

The authors have demonstrated what is the minimum sound to cause extra swimming in the ray. While I realize they are trying to argue that behavior experiments show greater sensitivity than AEPs, they cannot determine if the ray detected lower intensity sounds and did not respond. It is

similar to teleost predator prey studies. While you can determine the distance that fish reacted to the prey, you cannot be certain, it did not detect the prey at greater range but only reacted at shorter range to increase capture success. Considering, the authors indicate that the sound is “bothering” the rays, then one can imagine that “sub bothering” thresholds could result in detectiob but not in motor activity. I would suggest substituting a more scientific description than “bother”

“We also present clear evidence that the rays were bothered by increasing sound levels. An escape response was quantified when the stingray swam along the perimeter of the pen while flapping their pectoral fins; as this is not a common behaviour exhibited by wild stingrays, we hypothesize that the animals are trying to avoid or escape the sound. When exposed to lower frequencies, stingrays exhibited an increase in surface breach behaviour. While there is little data to explain this behaviour in stingrays, some animals jump or escape the water as a means of avoiding stressful conditions or to escape predators (Baylis 1982; Gibb et al. 2011; Mickle et al., 2018). Therefore, we also hypothesize that stingrays breach the water to avoid sounds”

The above paragraph is purely speculative and is not supported by data. Exactly what does “bothering” mean. While the authors did a credible job indicated that the stingrays increase their swimming during sound presentation, the other information in this paragraph is not supported. The rays are in an confined settings. Interactions with the perimeter or perhaps electric currents due to the presence of the rebar in the marine environment were not controlled for and cannot be discounted. True escape behaviors localize and move away from the stimulus and the experiment is not set up to test this. I routinely see pectoral fin flapping on the surface during sustained, straight line swimming events in Florida with rays when there is no evidence of a predator.

In the methods: the behavior was compared to one minute of “silence” prior to the sound turned on. What happened after the sound was turned off? Was the behavior sustained? Was five minutes a sufficient interval for the ray to regain normal behavior? Did they habituate to the sound? Perhaps they were irritated throughout the study resulting in greater movement. It is critical that the authors demonstrate that after the sound was stopped that the rays showed control behavior.

Minor

Opening sentence of abstract is strange. Most aquatic animals are hypothesized to detect sound over great distances because sound travels great distances. However, much of the shark work is focused on long range olfactory detection. I would suggest a more focused opening statement that does not detract from your study

Was there not a recent Australian study that concluded sound was not effective as a deterrent in sharks, this should be discussed

Perhaps this species has different behavior, but the coastal rays I have observed in Florida have extensive swimming bouts including pectoral fins out of the water. There was no predator in the water. Also, escape response usually mean M-cell mediated escape response in teleosts. Need to make this clear.

Are the tables formatted correctly. Seems an excessive number of vertical and horizontal lines
Figure 1: A diagram depicting a sound map of the experimental setup. Decibel levels were measured every meter along 4 transecting lines (including edges).

There must be something wrong with my figure as it not a sound map. It is a plain ellipse and the distances do not appear to be to scale. For example, the horizontal midline is 3.5 m on the left and ~2.5m on the right.

I strongly suggest that the authors show a representative sound map for the study

Figure 2: The hearing sensitivity of a) female and b) male southern stingrays using swimming activity as a marker of hearing. Significant differences between the sound levels are indicated by different letters. Error bars are representative of mean (+/- S.E)

Again, although the authors define hearing, it is not true hearing sensitivity. Rather it is the threshold that induces additional swimming. While I understand that the authors did not report true controls, they should at least compare this values to the 1 min control they did use

Fig 3 and 4

I would like to see the control values plotted. Also are not swimming and resting reciprocal? Therefore, resting may not need to be reported

(silent period between noise). I thought that only 1 min was being compared in this period

Perhaps this journal has really strange formatting rules, but Figure 3 and 4 legends look like multiple choice exam questions rather than professional figure legends

Author's Response to Decision Letter for (RSOS-191544.R0)

See Appendix B.

RSOS-191544.R1 (Revision)

Review form: Reviewer 1

Is the manuscript scientifically sound in its present form?

Yes

Are the interpretations and conclusions justified by the results?

Yes

Is the language acceptable?

Yes

Do you have any ethical concerns with this paper?

No

Have you any concerns about statistical analyses in this paper?

No

Recommendation?

Accept as is

Comments to the Author(s)

I am satisfied that the authors have made the corrections necessary to the manuscript to make it suitable for publication.

Review form: Reviewer 3

Is the manuscript scientifically sound in its present form?

Yes

Are the interpretations and conclusions justified by the results?

Yes

Is the language acceptable?

Yes

Do you have any ethical concerns with this paper?

No

Have you any concerns about statistical analyses in this paper?

No

Recommendation?

Accept as is

Comments to the Author(s)

I appreciate the authors' response to my suggestions. The manuscript is much clearer than the previous version. Also, the conclusions are more in line with the data presented.

Decision letter (RSOS-191544.R1)

17-Dec-2019

Dear Ms Mickle,

It is a pleasure to accept your manuscript entitled "Field assessment of behavioural responses of Southern Stingrays (*Hypanus americanus*) to acoustic stimuli" in its current form for publication in Royal Society Open Science. The comments of the reviewer(s) who reviewed your manuscript are included at the foot of this letter.

Please also send an alternative address for Rachel Pieniasek, as pieniasek@uwindsor.ca is not currently receiving messages from the journal.

on behalf of Dr Ryan Y Wong (Associate Editor) and Kevin Padian (Subject Editor)
openscience@royalsociety.org

Associate Editor Comments to Author (Dr Ryan Y Wong):
Dear Dr. Mickle,

Thank you for addressing the concerns of the reviewers. I am happy to announce that I can now recommend the paper be accepted for publication.

Reviewer comments to Author:
Reviewer: 1

Comments to the Author(s)
I am satisfied that the authors have made the corrections necessary to the manuscript to make it suitable for publication.

Reviewer: 3
Comments to the Author(s)
I appreciate the authors' response to my suggestions. The manuscript is much clearer than the previous version. Also, the conclusions are more in line with the data presented.

Appendix A

This topical paper has attempted to conduct behaviour audiograms on elasmobranchs. These studies are very difficult to perform in controlled tank environments, let alone in the field, which I commend the authors for doing. However, I do have a couple of major concerns, which could potentially be addressed by adding further details in the methods or discussion points. Hence my decision for major revision.

1. My first relates to the lack of detail in the methods regarding the behavioural paradigm employed. There is no information on where the animals were in the net pens when exposed to sound. Were the animals always at the same location? Or was this random? This leads into my next concern.

I do not think these can be classified as behavioural audiograms. To achieve this, these experiments need to be carefully controlled and the lack of detail in the methods regarding the behavioural paradigm prevents an understanding of exactly what was done. To generate a true behavioural audiogram the authors would need to condition the animal to move to the same spot in the net pen then expose them to calibrated tones. Therefore, the same propagation and acoustic conditions would exist for each trial. Plus, the authors do not actually present a "true" audiogram in the paper.

What the authors have done, was shown how loud specific tones need to be to elicit different types of escape responses. This has value in itself, but the tone of the paper needs to shift from behavioural audiograms to escape responses.

2. The 10 kHz signal to control for any potential responses to the electrical fields generated by the speakers is not correct. The electric field generated will oscillate at the frequency of the signal. For example, a 100 Hz tone will generate a 100 Hz oscillating electric field. Hence, the electric field generated at 10 kHz, will be way outside the frequency range the animals electrosensory system is sensitive too. Furthermore, the power to generate a 10 kHz signal is significantly less than at 100 Hz, therefore the size of the electric field at 10 kHz will be significantly smaller than at 100 Hz.
3. The final major point concerns the statistics. Along with the inconsistent presentation of p-values some of the significance's do not make sense. For example, line 32 (page 6) the authors state $F_{1, 29} = 12.482$ $p \leq 0.999$. Then on line 34 the authors state $F_{1, 29} = 12.482$ $p \leq 0.004$. How can one of these be significantly different and the other one not? When I look up the F Stat tables (old school) the F stat is significant, therefore I am not sure how they got that first p value. Also, the way the p value is written is incorrect this is suggesting that p is less than 0.9999, which obviously it is, the symbol should be reversed? There is multiple instances of this issue. Line 53 page 6 the authors state that swimming activity was

significantly different, but present $p=0.06$, which unless they are not using the normal significance level of $\alpha=0.05$ this is insignificant.

4. Given the random locations the rays could be in the net pen when the tones are turned on (that is my assumption given the lack of detail around this) there needs to be a sound map of the net pen.

Craig Radford

Appendix B

Field assessment of behavioural responses of Southern Stingrays (*Hypanus americanus*) to acoustic stimuli

135 Megan F. Mickle¹, Rachel H. Pieniazek¹, Dennis M. Higgs¹

¹Department of Biological Sciences, University of Windsor, 401 Sunset Avenue, Windsor, Ontario N9B 3P4, Canada

140 Rachel Pieniazek: pieniaz@uwindsor.ca

Dennis Higgs: dhiggs@uwindsor.ca

Corresponding author: Megan Mickle: micklem@uwindsor.ca 519-253-3000 x 4039

145 Keywords:

- Elasmobranchs
- Bioacoustics
- Behaviour
- Southern stingray
- 150 - Threshold
- Frequency
- Sound

155

160 Abstract:

The ability of elasmobranchs to detect and use sound cues has been heavily debated in previous research and has only recently received revived attention. To properly assess the importance of sound to elasmobranchs, ~~it is vital to~~ assessing their responses to acoustic stimuli in a field setting ~~is vital~~. Here, we establish a behavioural audiogram of free-swimming male and female southern stingrays (*Hypanus americanus*) exposed to low frequency tones. ~~We demonstrate that female stingrays exposed to tones (50-500 Hz) exhibit significant changes in swimming behaviours (increased time spent swimming, decreased rest time, increased surface breaches, and increased side swimming with pectoral flapping) at 140 dB re 1 μ Pa (-2.08 to -2.40 dB re 1 m s⁻²) while males exposed to the same tones did not exhibit a change in these behaviours until 160 dB re 1 μ Pa (-1.13 to -1.21 dB re 1 m s⁻²). Our results are the first demonstration of field responses to sound in the Batoidea and show a ~~clear ability to respond~~ distinct sensitivity to low frequency acoustic inputs.~~

170 Introduction:

Elasmobranchs possess multiple sensory and behavioural adaptations used for communication and migration, however, less is known about their hearing, mechanosensory systems, and functional use of the auditory system ~~as~~ compared to teleosts (Hueter et al., 2004). ~~Evidence suggests that elasmobranchs, e.g. lemon sharks (*Negaprion brevirostris*) and Atlantic stingrays (*Hypanus sabinus*), use their inner ears and lateral line to orient themselves to biotic sounds, such as prey items (Nelson & Gruber, 1963; Banner, 1968; Banner, 1972; Hodgson & Mathewson, 1978; Myrberg, 1969 & 1978 & 2001; Marusca & Tricas, 2004), suggesting that~~

sound detection may be important to the overall fitness of the animal. The hearing range of elasmobranchs studied to date falls within the range of 20 to 1000 Hz with greatest sensitivities at lower frequencies (Banner, 1968; Kelly & Nelson, 1975; Bullock et al., 1979; Casper & Mann, 2007; Casper & Mann, 2009). ~~Elasmobranch and teleost ears are both comprised of inner ear labyrinths containing a saccule, lagena, utricle, otoliths and three semicircular canals, however the elasmobranch ear is unique in that it also contains the macula neglecta (Lowenstein & Roberts, 1951, Popper & Fay, 1977; Tricas & New, 1997). The macula neglecta and sacculus are thought to be used for hearing in elasmobranchs (Lowenstein & Roberts, 1951, Popper & Fay, 1977).~~ Elasmobranch and teleost ears are both comprised of inner ear labyrinths containing a saccule, lagena, utricle, otoliths, and three semicircular canals, however the elasmobranch ear is unique in that it also contains the macula neglecta, which, combined with the sacculus, is thought to be used for hearing (Lowenstein & Roberts, 1951, Popper & Fay, 1977; Tricas & New, 1997)1977). Elasmobranchs detect sound~~and they can only detect sound~~ through particle motion (not pressure) because they lack a swim bladder and specialized hearing structures (Banner, 1967; Banner, 1972; Popper & Fay, 1973; Kelly & Nelson, 1975; Bullock & Corwin, 1979; Casper & Mann, 2006) which typically act as a pressure-to-displacement transducer organs in teleosts.

There is some evidence that elasmobranchs show an attraction response to low-frequency pulsed sounds (Nelson, 1967; Nelson et al., 1969; Myrberg et al., 1969; Nelson & Johnson, 1972), as they are thought to mimic stimuli produced by struggling prey (Myrberg, 2001). **Playback experiments using these pulsed sounds have been found to attract over 20 species of sharks, some from up to 100m from the initial sound source (Popper & Fay, 1977), although there has been criticism of this earlier work because sharks should only detect particle motion, which would not**

transmit far from the source (Gardiner et al., 2012; Casper & Mann, 2006, 2009). While there are a number of attraction experiments performed in elasmobranchs there is limited data is available regarding their potential adverse responses of sound on elasmobranchs. Klimley and Myrberg (1979) performed one of the few experiments establishing an avoidance response of sharks to sound in an aquarium setting, demonstrating that lemon sharks (*Negaprion brevirostris*) withdraw when presented with a broad band sound with a sudden onset of high intensity sound (20 dB above the ambient level). Recently, Chapius et al. (2019) showed that killer whale (*Orcinus orca*) calls and artificial sounds composed of mixed tones at frequencies from 20Hz to 10 kHz, caused a decrease in approaches of eight species of reef and coastal sharks to a baited camera setup. Although, Ryan et al. (2018) showed no effect of sound alone as a deterrent to feeding in Port Jackson (*Heterodontus portusjacksoni*) and epaulette (*Hemiscyllium ocellatum*) sharks in a laboratory setting. These Therefore, the mixed results sound as a deterrent, further show the need for enhanced field tests of behavioural responses of the hearing capabilities of elasmobranchs.

Elasmobranchs face several anthropogenic stressors such as: habitat degradation (Barnett et al., 2016), overfishing/bycatch (Bonfil, 1994; O'Connor et al., 2010; Jacoby et al., 2012; Bouyoucos et al., 2017), and provisioning (Semeniuk & Rothley, 2008; Brena et al., 2015) and are threatened in all the world's oceans (Dulvy et al. 2008, 2017). Rising levels of anthropogenic sounds are also increasingly recognized as a global threat to fishes (Slabbekoorn et al. 2010; Popper & Hawkins 2012) but, yet there are few studies conducted to date focusing on potential anthropogenic sound threats on elasmobranchs, and of the few studies, most focus on sharks with batoids poorly represented (Skomal & Mandelman, 2012). Here we exposed male and female

225 southern stingrays (*Hypanus americanus*) to low-frequency tones (50 to 1000 Hz) at differing
sound levels to create a behavioural audiogram of these animals. We show significant differences
in male and female responses to sound, with females responding to lower sound levels compared
to the males. to males, We this data can use this data can be further used to make hypotheses on
potential noise impacts on these animals stingrays.

230

Methods:

Capture and Transportation

Experiments were conducted following Canadian Council for Animal Care (CCAC) protocols
235 (University of Windsor AUPP 17-11). Experiments took place at the Bimini Biological Field
Station (BBFS) on the small island of South Bimini, Bahamas, which is small in size with
minimum commercial boating activity and some recreational power boats. Our study species was
the southern stingray, which is found in western Atlantic coastal waters, is a common benthic
mesopredator, located in western Atlantic coastal waters, with a diet composed mostly of
240 crustaceans and teleosts, and exhibits sexual dimorphism where females are on average are larger
than males (Gilliam & Sullivan, 1993; Henningsen & Leaf, 2010; Tilley & Strindberg, 2013;
Hayne et al., 2018). To catch stingrays, two sixteen-foot Sundance skiffs (Mercury Sea Pro
engine), equipped with two rubber dip nets (40" x 40"), four spoons (devices made of two PVCs
and plastic netting), and a plastic holding pool (approximately 4 x 4 x 2 ft), were driven to the

245 mangroves around the South Island of Bimini, during February-March 2019. Stingrays were often caught during mid-tide as this was the most efficient time to catch the animals. Once an animal was spotted, individuals (approximately 6 in total) got into the water to surround and capture the animal and transfer it to the holding tank on the skiff with the dip net. Once captured, animals were scanned using a PITtag reader (GPR Plus Reader, Biomark); if the stingray was
250 previously experimented on, it was released. If the animal did not have a PITtag number, they would be tagged following experimentation to avoid further stress to the animal prior to the study. Stingrays were then transported to a holding net pen (15 x 10 m) kept in the ocean at Bimini Biological Field Station (BBFS), with total travel times between 5 and 18 minutes from capture. Animals acclimated in the holding pen for approximately 24 to 40 hours prior to
255 experimentation. Stingrays were then identified again using the PITtag reader and transported from the holding pen to the experiment pen. Male stingrays caught ranged in total length from 44.6 to 115.2 cm while females ranged from 107-140 cm; there was no exclusion of stingrays caught based on their size.

Experiment

260 A circular experimental net pen (5 x 5 m), composed of metal rebar and plastic mesh netting extending underneath the sand, and mounted with two GoPro cameras (Hero 7) mount to the rebar, was created in the ocean (with a water depth ranging from 35-75 cm) to establish a behavioural audiogram of stingrays in both a control and noisy setting. During experimentation, outside temperatures ranged from 17-30° C, water temperature ranged from 22-23° C, water
265 depth ranged from x-ycm, and wave conditions ranged from 0-2 on the Beaufort scale.

Quantified behaviours included: time spent swimming, resting, side swimming (time spent

swimming vertically along the perimeter of the pen flapping pectoral fins) and surface breaches (head out of the water). As stingrays are generally sedentary animals (Tiley & Strindberg, 2013; Branco-Nunes et al., 2016) and ~~in our experiments~~ settled to the bottom after acclimation in our
270 experiments, an increase in swimming behaviour along the bottom of the pen was used as the prime metric for a threshold response to sound and increases in side swimming and breaching behaviours were used as metrics of an agitated response to the sound stimuli.

To perform sound experiments, two low-frequency underwater speakers (Clark Synthesis Diluvio AQ339; Lubell Labs) were placed adjacently along the perimeter of the pen, connected
275 to an amplifier (Scosche SA300), a 12 Volt PBS car battery, and an mp3 player (Sony Walkman NWZ-E464). Twenty stingrays (9 males and 11 females) were exposed to five low-frequency tones: 50, 90, 200, 500, and 1000 Hz, for these tones are hypothesized to overlap with their hearing range (Casper & Mann, 2009). The sequence of each frequency was randomized using a random number generator application to avoid pseudoreplication. Testing involved a 2-3 hour
280 acclimation period in the experimental net pen followed by sounds played in a stepwise pattern with a 1 min sound period followed by 5 mins of silence until all sounds were presented (Rollo & Higgs 2008; Isabella-Valenzi & Higgs 2013; Mickle et al., 2018). Stingrays were tested individually to avoid any follower bias. Behaviours during the 1 min sound treatment were directly compared with the 1 min of silence prior to sound, creating a difference metric. Sound
285 level was measured in pressure units (dB re 1 μ Pa) using a hydrophone (Inter Ocean system inc. – Acoustic Calibration and System Model 902), as well as acceleration units (dB re 1 $\text{m} \cdot \text{s}^{-2}$). Acceleration was estimated by the pressure gradient between hydrophone readings taken exactly 1m apart using the Euler equation (Mann 2006, Table 1). While we recognize that acceleration

units are the most relevant for detection at the level of the ray ear (Popper & Fay, 1973; Kelly & Nelson, 1975; Bullock & Corwin, 1979; Casper & Mann, 2006), we also provide pressure units to make it easier for other investigators measuring sound in the ocean, as underwater accelerometers remain difficult to obtain for open-field studies (Nedelec et al. 2017; Popper et al., 2019). A sound map was created to measure background sound levels along 27 locations of the net-pen using a hydrophone (Inter Ocean system inc. – Acoustic Calibration and System Model 902). The range of decibel levels for each frequency were also measured along 27 locations of the pen to establish a range of sound intensity the animals were exposed to during experimentation (Fig. 1; Table 2). Background sound levels were also measured in the middle of the pen prior to each experiment and ranged from 113 to 124 dB re 1 μ Pa. To establish the auditory threshold of stingrays, both males and females were first exposed to each frequency at 140 dB re 1 μ Pa (-2 dB re 1 $\text{m} \cdot \text{s}^{-2}$, Table 2). If the animal exhibited a change in swimming level, then decibel levels were decreased (by 5 dB) until the animal stopped exhibiting a response to the sound. If the animal did not alter their movement at 140 dB re 1 μ Pa, decibel levels were increased (by 10 dB) until a change in behaviour was noted. Decibel levels were increased by 10 instead of 5 dB as it was found that increases by increments of 5 dB re 1 μ Pa showed no change in behaviour. However, it is noteworthy to mention that there were three male stingrays exposed to 150 dB initially, due to time constraints based on decreasing tide. To help ensure animals were not responding to the speakers' baseline electrical output, we also played 10,000 Hz at 150 dB re 1 μ Pa (-1.6903 dB re 1 $\text{m} \cdot \text{s}^{-2}$) to each animal, although we do recognize they may have also responded to the speaker electrical output driven by the lower frequency sound output (i.e. a 100 Hz sound stimulus may also output a 100 Hz electrical output in addition to the speaker

background output). After each experiment, stingrays were sexed and measurements of disc length, disc width, spiracle width, and total length were taken (Table 3). Field work presents difficulties in terms of weather conditions, so to best control for this experimental trials were performed on days with similar weather, temperature conditions, and tide range. Behaviour
315 videos were analyzed using the software program “Soloman Coder” (Version: beta 19.08.02).

Statistical Analysis

Prior to the field season a power analysis was conducted to determine an appropriate sample size (8 stingrays of each sex) and to ensure the assumptions of normality would not be violated. Differences in stingray behaviour between the control and treatment periods were assessed using
320 repeated measures ANOVA for each metric: time spent swimming or resting, side swimming and surface breaches. To account for habituation or potential differences due to time spent in pen, behaviours were analyzed as a difference metric between the final minute of the five-minute control and the one-minute sound presentation. To create the difference metric, simple contrasts relative to 0 in the repeated measures design were implemented. A linear regression was
325 performed to determine if the size of the stingray and water depth had an effect on swimming activity of the fish. The regression was performed at the determined average threshold of the stingrays, 140 dB re 1 μ Pa for females and 160 dB re 1 μ Pa for males, at 50 Hz, as any potential influence of water depth would be greatest at the lowest frequency presented. Normality was tested using the descriptive statistics function in SPSS (IBM SPSS Statistics).

330 Results:

Both male and female southern stingrays exhibited a change in behaviour when exposed to low frequency tones. Females responded to tones at an average RMS of 140 dB re 1 μ Pa, while males responded to an average RMS of 160 dB re 1 μ Pa (see Table 1 for respective acceleration units throughout results).

335 *Threshold differences:*

We used swimming activity along the bottom of the pen as a marker of hearing to determine threshold. For females, there was an overall significant effect of sound level on swimming activity at 50, 90, 200, and 1000 Hz (Table 4, Fig. 2a). In subsequent post hoc analysis at these frequencies, there was no difference in swimming activity of female stingrays when comparing 340 130 and 135 dB re 1 μ Pa ($p > 0.999$ for all frequencies, Fig. 2a) but there was an increase in swimming activity at 140 dB re 1 μ Pa relative to the lower sound levels across all frequencies ($p < 0.004$, Fig. 2a).

There was an overall significant effect of sound level on swimming activity in males at 50, 90, 200 Hz (Table 4, Fig. 2b). Post hoc analyses indicate no difference in swimming activity of male 345 stingrays between 140 and 150 dB re 1 μ Pa ($p > 0.968$ for all frequencies, Fig. 2b) but there was a significant difference between 140 and 160 dB re 1 μ Pa, as males exhibited an increase in swimming activity at 50, 90, 200 Hz at 160 dB re 1 μ Pa ($p > 0.008$, Fig. 2b). Males also displayed an increase in swimming activity at 50 and 200 Hz at 160 dB re 1 μ Pa when compared to 150 dB re 1 μ Pa ($p > 0.001$, Fig. 2b).

350 *Frequency differences*

To examine a behavioural response to frequency, activity, resting and side swimming as well as number of surface breach events were measured at 140 dB re 1 uPa for females and 160 dB re 1 uPa for males, as these were both identified as threshold intensities for each sex. There was a significant effect of frequency on the swimming activity of both female ($F_{5,6}=10.935, p=0.006$, Fig. 3a) and male stingrays ($F_{5,3}=16.470, p=0.022$, Fig. 4a). Post-hoc analyses demonstrated significant increases in swimming activity of females relative to the control period at 50, 90, 200, and 1000 Hz while males were affected by 50, 90, 200, and 500 Hz. The number of surface breach events was also affected by sound exposure for both females ($F_{5,5}=1.932, p\leq 0.001$, Fig. 4c) and males ($F_{5,2}=16.578, p\leq 0.012$, Fig. 4c) with 50, 90, and 200 Hz tones showing a significant increase in breaching events compared to controls. Time spent resting in females ($F_{5,5}=3.291, p\leq 0.019$, Fig. 3b) and males ($F_{5,2}=2.226, p=0.050$, Fig. 4b) was also affected by frequency, causing a decrease in female resting rates when exposed to 50, 90, 200, and 500 Hz and male resting rates at 50, 90, and 200 Hz. Side swimming increased during sound exposure for both females ($F_{5,6}=7.237, p\leq 0.030$, Fig. 3d) and males ($F_{5,3}=3.418, p\leq 0.029$, Fig. 4d) at 50, 90, 200, and 500 Hz. Neither male (rest: $F_{1,8}=, p=0.347$; breach: $F_{1,8}=1.00, p=0.347$; swimming: $F_{1,6}=2.141, p=0.194$; side swimming: $F_{1,8}=0.885, p=0.374$) nor female (rest: $F_{1,10}=0.820, p=0.170$; breach: $F_{1,10}=1.00, p=0.341$; swimming: $F_{1,10}=0.188, p=0.674$; side swimming: $F_{1,10}=1.815, p=0.208$) stingrays responded to 10,000 Hz indicating that behavioural changes were true acoustic responses and not responses to speaker electrical output. The variations of water depth and total fish length did not have a significant effect on the activity levels of male (Adjusted $R^2 = -0.138$; $F_{1,7}=0.151, p=0.711$; Adjusted $R^2 = 0.192$; $F_{1,7}=2.428, p=0.180$, respectively) and female (Adjusted $R^2 = -0.029$; $F_{1,10}=0.721, p=0.418$; Adjusted $R^2 = -0.045$, $F_{1,10}=0.566, p=0.471$,

respectively) stingrays when exposed to 50 Hz at their average thresholds. Behaviour data were normally distributed.

375 Discussion

For the first time we quantify the behavioural thresholds of southern stingrays to a sound source and demonstrate that females respond to lower decibel levels (140 dB re 1 μ Pa; -2 dB re 1 $m \cdot s^{-2}$) than males (160 dB re 1 μ Pa; -1 to -1.2 dB re 1 $m \cdot s^{-2}$).

Threshold:

380 Southern stingrays are generally sedentary bottom-dwelling animals (Tilley & Strindberg, 2013; Branco-Nunes et al., 2016), therefore, we quantified resting behaviour as residing at the bottom of the pen without movement, while an increase in swimming along the bottom of the pen, indicated a response to sound. Stingrays were haphazardly distributed along the southeast quadrant of the net pen facing the open ocean (Fig 6) prior to experimentation, indicating that
385 stingrays were exposed to similar sound levels at the start of each experiment. As stingrays increased time spent swimming and decreased resting time during treatments ranging from 50-500 Hz, we conclude that sound elicited a change in normal stingray behaviour. While we did note that females still exhibited a response at 1000Hz, which was not detected in males, the swimming activity that was recorded were less than in the 50, 90, 200 and 500Hz.

390 Barber et al. (1985) discovered significant sex differences in the macula neglecta and ramus neglectus of the thornback ray (*Raja clavata*), with hair cell and axon number increasing with the size of the animal, and females have larger hair cell counts compared to similarly sized males. As

previously mentioned, the macula neglecta and sacculus are used primarily for hearing, therefore, gender differences may be involved in the location of prey, mate detection or other reproductive processes (Barber et al., 1984). Corwin et al. (1983) discovered that the number of hair cells in *R. clavata*, increases from 500 at birth to approximately 6000 in seven year old rays, further hypothesizing that the increase in hair cell counts is related to an increase in hearing sensitivity. For the current study, female stingrays had an average total length of 141.5 cm while the males were on average 88.9 cm (Table 3). As hair cell numbers in the macula neglecta increase with the size of the animal, the differences in hearing threshold observed in the current study may simply be explained by the size of macula neglecta and number of hair cells, however we did not see a significant effect of the size of the stingray on swimming activity levels of both male and females. It is also noteworthy to state that the size of the male and female rays were similar in the study by Barber et al. (1984) and there was still a significant difference in hair cell and axon number in the macula neglecta. Therefore, more research is needed to determine if differences are based on age, size or sex of the animals. Future studies should focus on similarly-sized male and female stingrays to determine if sex differences in hearing threshold are still present.

Frequency:

Our findings on frequency detection are consistent with previous research showing that elasmobranchs can hear from 20 to 1000 Hz with greatest sensitivities at lower frequencies (Banner, 1968; Kelly and Nelson, 1975; Bullock et al., 1979; Casper & Mann, 2007; Casper & Mann, 2009). Our behavioural evidence suggests that southern stingrays are most sensitive to frequencies of 50 up to 500 Hz, which is consistent with the results obtained from Corwin

415 (1983), demonstrating that the best hearing sensitivities of the thornback ray (*Raja clavata*) are between 40 and 200 Hz.

When exposed to lower frequencies, stingrays exhibited an increase in surface breach behaviour. While there is little data to explain this behaviour in stingrays, some animals jump or escape the water as a means of avoiding stressful conditions or to escape predators (Baylis 1982; Gibb et al. 420 2011; Mickle et al., 2018). Therefore, we also hypothesize that stingrays breach the water to avoid sounds.

425 *Future considerations and conclusions:*

For the first time, we show behavioural evidence of hearing in a stingray species, confirming that rays hear within the range of 50-1000 Hz, with greatest sensitivities at lower frequencies and that females respond to lower sound levels than males.

430 Recently, lab-based hearing studies have come under increasing scrutiny due to problematic acoustics in tank environments (Rogers et al., 2016; Campbell et al., 2019; Popper et al., 2019), suggesting an increased need for studies in a more natural setting. However, conducting acoustic studies in a field setting presents some challenges as field conditions cannot be as readily controlled as in a laboratory environment. For example, there was a variation in depth (35 to 75 cm), weather conditions (cloud cover) and wave action (0-2 on a beaufort scale) during

435 experimentation. As wild stingrays were caught for experimentation, they exhibited a variation in
size, and there is no way of knowing if they were exposed to anthropogenic sound prior to
capture. It is noteworthy to state that the stingrays may have detected the frequencies at lower
intensities of sound and did not respond, however without using an Auditory Evoked Potential
(AEP) technique, we cannot successfully examine this phenomenon. While underwater acoustics
440 are never perfect, and we still see a use for laboratory-based experiments, our approach offers a
promising avenue for continued investigation of auditory responses in elasmobranchs.

The behavioural responses we see to low-frequency sounds, and especially the increasing surface
breach behaviours to louder sounds, is interesting in its own right, but also calls for concerns for
these animals exposed to anthropogenic sounds. The increasing anthropogenic cacophony of the
445 underwater environment is of significant concern worldwide and has already been shown to have
important fitness consequences for teleosts (Popper and Hastings, 2009; Slabbekoorn et al.,
2010; Radford et al., 2014; Mickle and Higgs, 2017). Despite the increasing concern for
anthropogenic sound as an ecological stressor, we have no knowledge of these possible effects
on elasmobranchs, one of the most imperiled groups of fish worldwide (Dulvy et al., 2008;
450 Barnett et al., 2016). Elasmobranchs are important predators that often face anthropogenic
threats, with approximately 1.5 million tons of elasmobranchs killed annually as bycatch (Bonfil,
1994; O'connor et al., 2010; Jacoby et al., 2012) and despite declines in elasmobranch
populations (Dulvy et al., 2008), there is little research regarding potential sound stressors within
their environment, which can be important to help conserve these species. Next steps of this
455 research should include quantifying the effects of sound as a potential stressor on this species
and to expand the current approach to other elasmobranchs.

Acknowledgements:

We would like to thank Kirsten Poling, Christina Semeniuk and Nigel Hussey for assistance with designing this project. We would also like to thank Bimini Biological Field Station for assistance with stingray capturing and data collection. This research was funded by the Natural Sciences and Engineering Research Council (NSERC).

465

Table Legend

Table 1: Sound level measured as acceleration units (dB re 1 m/s²). Acceleration was estimated by the pressure gradient between hydrophone readings taken exactly 1m apart using the Euler equation (Mann 2006).

470 **Table 2:** Average decibel levels of each frequency along 27 locations (see Fig. 1) of the net-pen to establish a range of sound intensity the animals were exposed to during experimentation.

Table 3: The range of measurements of both male and female stingrays used in experiments.

Table 4: Statistical representation of main effects of frequencies (50-1000Hz) at 140 dB re 1 μ Pa for females and 160 dB re 1 μ Pa for males, on swimming activity levels.

475

Figure Legend

Figure 1: A soundmap showing sound levels (dB 1 μ Pa) across the experimental pen for sound presentations of a) 50 Hz at 140 dB re 1 μ Pa b) 500 Hz at 140 dB re 1 μ Pa c) 50 Hz at 160 dB re 1 μ Pa d) b) 500 Hz at 160 dB re 1 μ Pa. Sound level was measured at 27 locations in the pen,

480 with intermediate levels interpolated between recording locations to represent sound level as a
heat map across the entire experimental arena.

Figure 2: The hearing threshold of a) female and b) male southern stingrays using swimming activity as a marker of hearing. Significant differences between the sound levels are indicated by different letters. Error bars are representative of mean (+/- S.E).

485 **Figure 3:** Female stingray behaviour in response to low frequency tones at 140 dB re 1 μ Pa, relative to controls (silent period between sound) a) mean swimming activity b) mean resting activity c) mean surface breach events d) mean time spent side swimming. Error bars are representative of mean (+/- S.E). Significant differences relative to control are indicated by an * while differences compared to other frequencies are indicated by different letters.

490 **Figure 4:** Male stingray behaviour in response to low frequency tones at 160 dB re 1 μ Pa, relative to controls (silent period between sound). a) mean swimming activity b) mean resting activity c) mean surface breach events d) mean time side swimming. Error bars are representative of mean (+/- S.E). Significant differences relative to control are indicated by an * while differences compared to other frequencies are indicated by different letters.

495

Appendix

500 Table 1

	Decibel level (dB re 1 μ Pa)				
Frequency (Hz)	130	135	140	150	160
50	-2.93292	-2.47048083	-2.40206	-1.61007	-1.20703
90	-2.75886	-2.44377776	-2.11007	-1.46007	-1.21509
200	-2.85886	-2.37562758	-2.12494	-1.59378	-1.07494
500	-2.70703	-2.31006671	-2.08504	-1.35169	-1.1283
1000	-2.75703	-2.41508854	-2.07563	-1.85703	-1.12563

Table 2

Decibel levels (dB re 1 μ Pa)	50 Hz	90 Hz	200 Hz	500 Hz	1000 Hz
130	131.2	130.2	130.7	129.8	129.5
135	134.6	136.7	135.4	134.2	136.1
140	137.3	141.5	143.1	143.3	140.2
150	147.5	145.1	146.3	149.3	148.3
160	156.1	157.8	155.5	156.3	153.4

Table 3

Measurements (cm)	Female	Male
Total length	107 – 140	44.6 – 115.2
Disc width	59.2 – 91	42.9 – 55.2
Disc length	48 – 74	35.7 – 43.4
Spiracle width	9 – 15.2	6.9 – 9.9
Barb length	9.8 – 10.8	5.5 – 9.9

505

Table 4

Frequency (Hz)	Female		Male	
	F statistic	P value	F statistic	P value
50	$F_{3,29} = 10.803$	$p < 0.001$	$F_{2,21} = 20.331$	$p < 0.001$
90	$F_{3,29} = 9.218$	$p < 0.001$	$F_{2,21} = 5.604$	$p = 0.011$
200	$F_{3,29} = 5.684$	$p = 0.003$	$F_{2,21} = 26.052$	$p < 0.001$
500	$F_{3,29} = 3.826$	$p = 0.0120$	$F_{2,21} = 3.673$	$p = 0.043$
1000	$F_{3,29} = 1.392$	$p = 0.265$	$F_{2,21} = 0.165$	$p = 0.849$

510

515

520

525

Work cited

Banner, A. (1968). Attraction of young lemon sharks, *Negaprion brevirostris*, by sound. *Copeia*, 1968(4), 871-872. Doi:10.2307/1441861.

530 Banner, A. (1972). Use of sound in predation by young lemon sharks, *Negaprion brevirostris*. *Bulletin of Marine Science*, 22(2), 251-283.

Barber, V. C., Yake, K. I., Clark, V. F., & Pungur, J. (1985). Quantitative analyses of sex and size differences in the macula neglecta and ramus neglectus in the inner ear of the skate, *Raja ocellata*. *Cell and tissue research*, 241(3), 597-605.

535 Barnett, A., Payne, N. L., Semmens, J. M., & Fitzpatrick, R. (2016). Ecotourism increases the field metabolic rate of whitetip reef sharks. *Biological Conservation*, 199, 132-136. Doi: 10.1016/j.biocon.2016.05.009.

Baylis, J.R. (1982). Unusual escape response by two cyprinodontiform fishes, and a bluegill predator's counter-strategy. *Copeia*. No. 2.

540 Bouyoucos, I.A., Suskia, C.D., Mandelmanc, J.W. & Brookes, J.W. (2017). The energetic, physiological, and behavioral response of lemon sharks (*Negaprion brevirostris*) to simulated

longline capture. *Comparative Biochemistry and Physiology Part A: Molecular and Integrative Physiology*, 207, 65-72. Doi:10.1016/j.cbpa.2017.02.023.

Bonfil, R. (1994). Overview of world elasmobranch fisheries (No. 341). Food & Agriculture Org.

545 Branco-Nunes, I., Veras, D., Oliveira, P., & Hazin, F. (2016). Vertical movements of the southern stingray, *Dasyatis americana* (Hildebrand & Schroeder, 1928) in the Biological Reserve of the Rocas Atoll, Brazil. *Latin American Journal of Aquatic Research*, 44(2), 216-227. Doi: 10.3856/vol44-issue2-fulltext-3.

550 Brena, P. F., Mourier, J., Planes, S., & Clua, E. (2015). Shark and ray provisioning: functional insights into behavioral, ecological and physiological responses across multiple scales. *Marine Ecology Progress Series*, 538, 273-283. Doi: 10.3354/meps11492.

Bullock, T. H., & Corwin, J. T. (1979). Acoustic evoked activity in the brain in sharks. *Journal of comparative physiology*, 129(3), 223-234. Doi: 10.1007/BF00657658.

555 Campbell, J. (2019). Particle motion and sound pressure in fish tanks: A behavioural exploration of acoustic sensitivity in the zebrafish, *Behavioural Processes*, 164, 38-47. Doi: 10.1016/j.beproc.2019.04.001.

Casper, B. M., & Mann, D. A. (2007). Dipole hearing measurements in elasmobranch fishes. *Journal of Experimental Biology*, 210(1), 75-81. Doi: 10.1242/jeb.02617

560 Casper, B. M., & Mann, D. A. (2009). Field hearing measurements of the Atlantic sharpnose shark *Rhizoprionodon terraenovae*. *Journal of fish biology*, 75(10), 2768-2776. Doi: 10.1111/j.1095-8649.2009.02477.x

565 Chapuis, L., Collin, S. P., Yopak, K. E., McCauley, R. D., Kempster, R. M., Ryan, L. A., Schmidt, C., Kerr, C.C., Gennari, E., Edgeberg, C.A., & Hart, N. S. (2019). The effect of underwater sounds on shark behaviour. *Scientific reports*, 9(1), 6924. Doi: 10.1038/s41598-019-43078-w

Dulvy, N. K., Baum, J. K., Clarke, S., Compagno, L. J., Cortés, E., Domingo, A., ... & Martínez, J. (2008). You can swim but you can't hide: the global status and conservation of oceanic pelagic sharks and rays. *Aquatic Conservation: Marine and Freshwater Ecosystems*, 18(5), 459-482. Doi:10.1002/aqc.975

570 Dulvy, N. K., Simpfendorfer, C. A., Davidson, L. N., Fordham, S. V., Bräutigam, A., Sant, G., & Welch, D. J. (2017). Challenges and priorities in shark and ray conservation. *Current Biology*, 27(11), 565-572. Doi: 10.1016/j.cub.2017.04.038.

575 Gardiner, J. M., Hueter, R. E., Maruska, K. P., Sisneros, J. A., Casper, B. M., Mann, D. A., & Demski, L. S. (2012). Sensory physiology and behavior of elasmobranchs. *Biology of sharks and their relatives*, 1, 349-401

- Gibb, A. C., Ashley-Ross, M. A., Pace, C. M., & Long, J. H. 2011. Fish out of water: terrestrial jumping by fully aquatic fishes. *Journal of Experimental Zoology Part A*, 315(10), 649-653. Doi: 10.1002/jez.711
- 580 Gilliam, D. S., & Sullivan, K. M. (1993). Diet and feeding habits of the southern stingray *Dasyatis americana* in the central Bahamas. *Bulletin of Marine Science*, 52(3), 1000-1007.
- Henningsen, A. D., & Leaf, R. T. (2010). Observations on the captive biology of the southern stingray. *Transactions of the American Fisheries Society*, 139(3), 783-791. Doi: 10.1577/T09-124.1
- 585 Hodgson, E. S., & Mathewson, R. F. (1978). Sensory biology of sharks, skates, and rays. Tufts University Medford, Department of Biology.
- Hueter, R. E., Mann, D. A., Maruska, K. P., Sisneros, J. A., & Demski, L. S. (2004). Sensory biology of elasmobranchs. *Biology of sharks and their relatives*, 325-368.
- 590 Jacoby, D. M., Croft, D. P., & Sims, D. W. (2012). Social behaviour in sharks and rays: analysis, patterns and implications for conservation. *Fish and Fisheries*, 13(4), 399-417. Doi: 10.1111/j.1467-2979.2011.00436.x
- Jordan, L.K., Mandelman, J.W., McComb, M., Fordham, S.V., Carlson, J.K and Werner, T.B. (2013). Linking sensory biology and fisheries bycatch reduction in elasmobranch fishes: a review with new directions for research. *Conservation Physiology*, (1), 1-20. Doi: 10.1093/conphys/cot002.
- 595 Kalmijn, A. J. (1982). Electric and magnetic field detection in elasmobranch fishes. *Science*, 218(4575), 916-918. Doi: 10.1126/science.7134985.
- Kelly, J. C., & Nelson, D. R. (1975). Hearing thresholds of the horn shark, *Heterodontus francisci*. *The Journal of the Acoustical Society of America*, 58(4), 905-909.
- 600 Klimley, A. P., & Myrberg Jr, A. A. (1979). Acoustic stimuli underlying withdrawal from a sound source by adult lemon sharks, *Negaprion brevirostris*. *Bulletin of Marine Science*, 29(4), 447-458.
- 605 Lowenstein, O., & Roberts, T. D. M. (1951). The localization and analysis of the responses to vibration from the isolated elasmobranch labyrinth. A contribution to the problem of the evolution of hearing in vertebrates. *The Journal of Physiology*, 114(4), 471-489.
- Mann, D. A. (2006). Propagation of fish sounds. In *Communication in Fishes* (ed. F. Ladich, S. P. Collin, P. Moller and B. G. Kapoor), pp. 107-120. Enfield, NH: Science Publishers.
- 610 Maruska, K.P. and Tricas, T.C. (2004). Test of the mechanotactile hypothesis: neuromast morphology and response dynamics of mechanosensory lateral line primary afferents in the stingray. *Journal of Experimental Biology*, (207), 3463-3476. Doi: 10.1242/jeb.01140.

- Mickle, M. F., & Higgs, D. M. (2017). Integrating techniques: a review of the effects of anthropogenic sound on freshwater fish. *Canadian Journal of Fisheries and Aquatic Sciences*, (999), 1-8. Doi: 10.1139/cjfas-2017-0245.
- 615 Mickle, M. F., Miehl, S. M., Johnson, N. S., & Higgs, D. M. (2018). Hearing capabilities and behavioural response of sea lamprey (*Petromyzon marinus*) to low-frequency sounds. *Canadian Journal of Fisheries and Aquatic Sciences*, (999), 1-8.
- Myrberg, A. A., Banner, A., & Richard, J. D. (1969). Shark attraction using a video-acoustic system. *Marine Biology*, 2(3), 264-276. Doi: 10.1007/BF00351149
- 620 Myrberg, A. A. (1978). Underwater sound-its effect on the behavior of sharks. *Sensory biology of sharks, skates and rays*, 391-417.
- Myrberg, A. A. (2001). The acoustical biology of elasmobranchs. In *The behavior and sensory biology of elasmobranch fishes: an anthology in memory of Donald Richard Nelson* (pp. 31-46). Springer, Dordrecht.
- 625 Nelson, D. R., & Gruber, S. H. (1963). Sharks: attraction by low-frequency sounds. *Science*, 142(3594), 975-977.
- Nelson, D. R. (1967). Hearing thresholds, frequency discrimination, and acoustic orientation in the lemon shark, *Negaprion brevirostris*. *Bulletin of Marine Science*, 17(3), 741-768.
- 630 Nelson, D. R., Johnson, R. H., & Waldrop, L. G. (1969). Responses in Bahamian Sharks and Groupers, to Low-Frequency, Pulsed Sounds. *Bulletin of the Southern California Academy of Sciences*, 68(3), 131-137.
- Nelson, D. R., & Johnson, R. H. (1972). Acoustic attraction of Pacific reef sharks: effect of pulse intermittency and variability. *Comparative Biochemistry and Physiology Part A: Physiology*, 42(1), 85-95. Doi: 10.1016/0300-9629(72)90370-2.
- 635 O'Connell, C. P., Abel, D. C., Rice, P. H., Stroud, E. M., & Simuro, N. C. (2010). Responses of the southern stingray (*Dasyatis americana*) and the nurse shark (*Ginglymostoma cirratum*) to permanent magnets. *Marine and Freshwater Behaviour and Physiology*, 43(1), 63-73. Doi: 10.1080/10236241003672230.
- 640 Popper, A. N., & Fay, R. R. (1973). Sound detection and processing by teleost fishes: a critical review. *The Journal of the Acoustical Society of America*, 53(6), 1515-1529. Doi: 10.1159/000113821.
- Popper, A. N., & Fay, R. R. (1977). Structure and function of the elasmobranch auditory system. *American Zoologist*, 17(2), 443-452.
- Popper, A.N., Hawkins, A.D., Sand, O., & Sisneros J.A. (2019). Examining the hearing abilities of fishes. *Journal of the Acoustical Society of America*, 146 (2), 948-955.

- 645 Radford, A.N., Kerridge, E., and Simpson, S.D. (2014). Acoustic communication in a noisy world: can fish compete with anthropogenic noise? *Behavioural Ecology*, 25(5), 1022-1030. Doi: 10.1093/beheco/aru029.
- Robins, C.R. & G.C. Ray. 1986. Peterson Field Guide: Atlantic Coast Fishes. Houghton Mifflin Company, New York, NY, USA, 354 p.
- 650 Rogers, P. H., Hawkins, A. D., Popper, A. N., Fay, R. R., and Gray, M. D. (2016). "Parvulescu revisited: Small tank acoustics for bioacousticians," in *The Effects of Noise on Aquatic Life II*, edited by A. N. Popper and A. D. Hawkins (Springer Science Business Media, New York), pp. 933–941. Doi: 10.1007/978-1-4939-2981-8_115.
- 655 Ross, Q. E., Dunning, D. J., Thorne, R., Menezes, J. K., Tiller, G. W., & Watson, J. K. (1993). Response of alewives to high-frequency sound at a power plant intake on Lake Ontario. *North American Journal of Fisheries Management*, 13(2), 291-303.
- Semeniuk, C. A., & Rothley, K. D. (2008). Costs of group-living for a normally solitary forager: effects of provisioning tourism on southern stingrays *Dasyatis americana*. *Marine Ecology Progress Series*, 357, 271-282. Doi: 10.3354/meps07299.
- 660 Sisneros, J. A., & Tricas, T. C. (2002). Neuroethology and life history adaptations of the elasmobranch electric sense. *Journal of Physiology-Paris*, 96(5-6), 379-389. Doi: 10.1016/S0928-4257(03)00016-0.
- 665 Skomal, G. B., & Mandelman, J. W. (2012). The physiological response to anthropogenic stressors in marine elasmobranch fishes: a review with a focus on the secondary response. *Comparative Biochemistry and Physiology Part A: Molecular & Integrative Physiology*, 162(2), 146-155. Doi: 10.1016/j.cbpa.2011.10.002.
- Slabbekoorn, H., Bouton, N., van Opzeeland, I., Coers, A., ten Cate, C., and Popper, A. N. 2010. A noisy spring: the impact of globally rising underwater sound levels on fish. *Trends in Ecology and Evolution*, 25(7), 419-427. Doi: 10.1016/j.tree.2010.04.005.
- 670 Tilley, A., & Strindberg, S. (2013). Population density estimation of southern stingrays *Dasyatis americana* on a Caribbean atoll using distance sampling. *Aquatic Conservation: Marine and Freshwater Ecosystems*, 23(2), 202-209. Doi: 10.1002/aqc.2317.
- 675 Tricas, T. C., & New, J. G. (1998). Sensitivity and response dynamics of elasmobranch electrosensory primary afferent neurons to near threshold fields. *Journal of Comparative Physiology A*, 182(1), 89-101.

680

685

690

Supplemental Information

Figure 5: A visual representation of the frequency range (50, 90, 200, 500, 1000 Hz) and relative amplitude (dB) of each tone played to the stingrays during experimentation. Recordings were taken with a hydrophone (Loggerhead Instruments, Model # HTI-96-Min/3V/Exp/LED).